# Enhancing Knowledge Transfer for Task Incremental Learning with Data-free Subnetwork

**Qiang Gao**[1,*], **Xiaojun Shan**[2,*], **Yuchen Zhang**[2], **Fan Zhou**[2,†]

[1]Southwestern University of Finance and Economics, Chengdu, China
[2]University of Electronic Science and Technology of China, Chengdu, China
qianggao@swufe.edu.cn, {xiaojunshan@std., yuchenzhang@std., fan.zhou@}uestc.edu.cn

## Abstract

As there exist competitive subnetworks within a dense network in concert with *Lottery Ticket Hypothesis*, we introduce a novel neuron-wise task incremental learning method, namely *Data-free Subnetworks (DSN)*, which attempts to enhance the elastic knowledge transfer across the tasks that sequentially arrive. Specifically, DSN primarily seeks to transfer knowledge to the new coming task from the learned tasks by selecting the affiliated weights of a small set of neurons to be activated, including the reused neurons from prior tasks via neuron-wise masks. And it also transfers possibly valuable knowledge to the earlier tasks via data-free replay. Especially, DSN inherently relieves the catastrophic forgetting and the unavailability of past data or possible privacy concerns. The comprehensive experiments conducted on four benchmark datasets demonstrate the effectiveness of the proposed DSN in the context of task-incremental learning by comparing it to several state-of-the-art baselines. In particular, DSN enables the knowledge transfer to the earlier tasks, which is often overlooked by prior efforts.

## 1   Introduction

Continual learning (CL), as one of the human-like lifelong learning or incremental learning paradigms, has received enormous attention in Artificial Intelligence (AI) community due to its capability to incrementally learn a sequence of tasks in a (deep) neural network and to keep accumulating knowledge throughout its lifetime [1]. According to the context of whether the task identity is provided during model training and inference, the majority of prior works categorize continual learning into three main practical problems by specific settings, including task-incremental learning, class-incremental learning, and task-free incremental learning. This study was prepared for Task Incremental Learning or TIL for short. Not limited to TIL, existing solutions for incremental learning are put forward to solve the catastrophic forgetting (CF) issue [2] that is a common phenomenon in CL scenarios. Usually, these approaches can be broadly categorized into three directions, i.e., regularization-based [3, 4, 5], rehearsal-based [6, 7, 8], and architecture-based [9, 10, 11, 12].

**Motivations:** Recently, several studies have demonstrated that deep neural networks are usually over-parameterized, whereby the redundant or unuseful weights can be pruned, allowing efficient computation and on-par or even better performance [13, 14, 15]. Likewise, researchers stated this phenomenon as *Lottery Ticket Hypothesis (LTH) [16, 17]*, i.e., *a randomly-initialized neural network contains a subnetwork such that, when trained in isolation, can match the performance of the original network.* With this in mind, more recent efforts turn to randomly initialize a large but sparse neural network and employ it to sequentially discover compact subnetworks for (task) incremental learning,

---

*Equal contribution.

†Corresponding author (`fan.zhou@uestc.edu.cn`).

aiming at providing room to learn new tasks as well as enabling forward knowledge transfer from the prior tasks [18, 19, 20]. Specifically, they concentrate on weight-wise search using an adaptive binary mask that determines which weight should be activated (*winning tickets*). Nevertheless, we consider that it is not preferable to maintain such a binary mask for retrieving a subnetwork, especially in incremental learning. Since it will destroy the continuous structure of a layer-dependent subnetwork. Meanwhile, maintaining a binary mask whose size equals the number of parameters also results in significant resource consumption. In addition to this, knowledge transfer across different tasks is the core work of incremental learning. Existing solutions primarily seek to enable the knowledge transfer from earlier tasks to the new coming task (i.e., *forward knowledge transfer*), along with considering the CF problem [21, 22, 23]. Like human cognitive processes [24], we debate whether the newly learned knowledge has the potential to help improve past tasks (i.e., *backward knowledge transfer*). Intuitively, it is natural and practical that we can replay the old samples from earlier tasks, like rehearsal-based approaches, to trigger the ability to tackle old tasks much better. However, this is not applicable in the actual incremental learning scenario due to computational overhead and privacy concerns. With those in mind, we ask a much more ambitious question: *Does LTH hold in the setting of TIL concerning elastic knowledge transfer when different tasks arrive sequentially?*

**Contributions:** To answer this question, we propose a novel and practical approach named Data-free Subnetworks (DSN), enabling the elastic knowledge transfer across the tasks that sequentially arrive. Motivated by LTH, we assume that there exists a hypernetwork (Fig. 1 (a)) that contains a sequence of competitive "ticket" sub-networks, where each of them can perform well on the affiliated task with the knowledge transfer in mind. DSN mainly contains two components, i.e., neuron-wise mask and data-free memory reply. Specifically, we first randomly initialize a hypernetwork and incrementally learn the model parameters and task-specific masks, where each mask determines which neurons and their corresponding weights should be used for a new coming task (Fig. 1 (b)). To allow the forward knowledge transfer on a new task, we use the mask to adaptively select the neurons that have been used in the earlier tasks and make their corresponding weights unchangeable for the purpose of addressing the CF problem. To permit the backward knowledge transfer, we first measure the mask similarity scores and craft the impressions of the most similar task via data-free memory replay. In contrast to sample replay paradigms, we treat the subnetwork as a past experience and use it to recall past impressions, avoiding incrementally preserving the past actual samples. Then, we fine-tune the most similar task and make backward knowledge transfer possible(Fig. 1 (c)). We summarize our contributions as follows:

- In concert with Lottery Ticket Hypothesis (LTH), we introduce a novel and practical task-incremental learning solution DSN, which aims to learn a compact subnetwork (winning ticket) for each task from the perspective of knowledge transfer, including forward and backward transfer.

- We devise a neuron-wise mask mechanism to adaptive select neuron-affiliated weights to transfer the learned knowledge to the new task, where the used neuron-affiliated weights in the past tasks are frozen to eliminate the CF problem. Besides, our proposed data-free replay mechanism regards the trained subnetwork as a past experience and uses it to craft impressions regarding the past samples, which does not require holding any actual samples related to past tasks.

- The comprehensive experiments conducted on four benchmark datasets demonstrate the effectiveness of our proposed DSN against the state-of-the-art baselines.

## 2 Related Work

**Catastrophic Forgetting.** Existing approaches to address the CF problem can be broadly categorized into three directions. *(1) Regularization-based approaches* that add a selective penalty in the loss function, which penalizes the variation of network parameters depending on its importance in performing previous tasks [3, 25, 4, 26, 27]. EWC operates the Fisher information matrix to estimate the importance weights [3]. Synaptic Intelligence (SI) provides an online approximation of parameter importance by assessing their contributions to the variations of total loss [28]. *(2) Rehearsal-based approaches* typically incrementally hold a few representative old training samples as a small memory buffer and replay it to retain the past knowledge when learning a new task [6, 7, 8, 29, 30, 31]. For example, GEM proposes to further guarantee an equal number of old training samples per class [32]. DGR presents a framework that replays generated data sampled from the old generative model while learning each generation task to inherit the previous knowledge [6]. Also, other

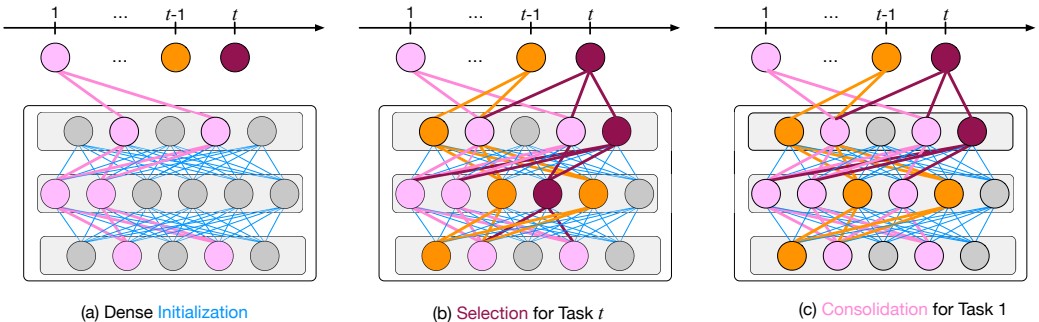

Figure 1: Task Incremental Learning with DSN: (a) We define the task network as a hypernetwork that can maintain the training of multiple tasks. And it will be randomly initialized (blue line) for task 1 training (pink circle and line); (b) We use the Neuron-wise Mask to produce a subnetwork for task $t$ training (purple circle and line); (c) If task 1 is the most similar to task $t$ by similarity measurement of masks, we update the architecture of task 1 based on the performance of Data-free Replay.

researchers [33, 34, 35] perform data-free generative replay to alleviate memory space or privacy concerns. For instance, [34] employs two teacher models to enhance the quality of generated samples. However, it requires an additional generative model, yielding additional trainable parameters. [35] can generate old samples merely through the main network. Besides, [33, 34, 35], which directly employ synthesized samples to fine-tune the model, have limitations in achieving forget-free performance and are susceptible to interference from other tasks. *(3) Architecture-based approaches* assign different model parameters to different tasks, freeze the parameters for old tasks by masking them out when training the new task, and expand the network when necessary [10, 11, 12, 8, 19]. DEN [36] can dynamically adjust its network capacity as it trains on a sequence of tasks to learn a compact overlapping knowledge-sharing structure among tasks. WSN aims to explore the subnetworks sequentially for a sequence of tasks from a sparse neural network [19]. It also maintains a binary mask whose size equals the number of parameters, resulting in significant resource consumption. We argue the architecture-based approaches provide us with the possibility to explore better subnetworks while they either confront the redundant neurons (weights) or shallow knowledge transfer.

**Knowledge Transfer.** Recently, researchers and practitioners have argued that the ability to learn from a sequence of tasks is in essence to complement and reinforce each other through knowledge transfer. Several studies have proposed to improve task performance by maximizing shareable knowledge [37, 38, 39, 40]. For instance, Bayes model [41, 42] and regression method [43] concentrate on transferring the learned knowledge to facilitate learning in the new domains or new tasks. BNS [12] presents a Task-CL algorithm that can adaptively determine the network structure for each new task and selectively transfer previously learned knowledge. However, it employs massive past samples to address the negative knowledge transfer concern. HAT [44] and WSN [19] selectively reuse the optimized parameters via a binary masking mechanism, enabling significant forward knowledge transfer. However, backward knowledge transfer is not considered. [45] focuses on a few-shot scene, which is accomplished by jointly learning model weights and adaptive non-binary soft masks, with the major subnetwork minimizing catastrophic forgetting and the minor subnetwork preventing overfitting. Also, it fails to take backward knowledge transfer into consideration. [46] proposes two distillation-based objectives for class incremental learning. This method can utilize the feature space structure of the previous model to preserve the representation of previous classes. Besides, a controlled transfer is introduced to approximate and condition the current model on the semantic similarities between incoming and prior classes, maximizing positive forward transfer and minimizing negative backward transfer. However, it can not achieve a positive backward transfer. [38] and [47] preserve previous data as a replay buffer for retraining old tasks, which raises concerns regarding data privacy and memory overhead. [38] utilizes a CL-plugin to explicitly identify and transfer useful features/knowledge from previous tasks to new ones, designed specifically for LLMs. [48] characterizes task correlations, identifies positively correlated old tasks, and selectively modifies the learned model of the old tasks when learning new tasks. In sum, the above solutions concentrate on either addressing severe negative transfer using pruning methods or using additional data to help knowledge consolidation. In contrast, our DSN attempts to select subnetworks from a sparse network for a sequence of tasks while using a data-free mechanism to enable the possibly positive knowledge transfer to the old tasks.

## 3 The Proposed Method

### 3.1 Problem Statement

Without loss of generality, a classical TIL scenario, typically in a supervised learning setting (e.g., classification), deals with a set $D = \{D_1, D_2, \cdots, D_T\}$ of $T$ tasks that come in turn to the learner or task network (e.g., a deep neural network). Let $D_t = \{(x_t^i, y_t^i)\}_{i=1}^{n_t}$ denote the dataset of task $t$, containing $n_t$ pairs of raw instances and independent label space. In our context, assume that we have a neural network, named hypernetwork $\mathcal{H}$. $\mathcal{H}(\cdot, \boldsymbol{\theta}(\boldsymbol{n}))$ is composed of layered neurons $\boldsymbol{n}$ where learnable weights $\boldsymbol{\theta}$ bridge them together. TIL follows the common continual learning paradigm and attempts to sequentially learn a set of task $T$ with the following objective for each task $t$:

$$\boldsymbol{\theta}^* = \underset{\boldsymbol{\theta}}{\text{minimize}} \frac{1}{n_t} \sum_{i=1}^{n_t} \mathcal{L}(\mathcal{H}(x_t^i; \boldsymbol{\theta}), y_t^i), \tag{1}$$

where $\mathcal{L}(\cdot, \cdot)$ is a loss function such as cross-entropy loss. It is important to note that $D_t$ for task $t$ is only accessible during task $t$ learning, and any instances in $D_t$ cannot be accessed while learning future tasks. Furthermore, assume that task identity is given in both the training and inference processes in the TIL scenario. In addition, we follow most of the previous studies and employ multi-head settings for TIL, with each task having its own single but non-overlapping head.

The Lottery Ticket Hypothesis (LTH) hypothesizes the existence of competitive subnetworks in a randomly initialized dense neural network [16], which motivates us to ingeniously devise an incremental learner that employs an over-parameterized deep neural network to allow more room for future task learning. Therefore, our goal is to discover an optimal subnetwork along with a neuron-wise mask $\boldsymbol{m}_t \in \{0, 1\}^{|\boldsymbol{n}|}$ for each task $t$ such that $|\boldsymbol{m}_t^*| \ll |\boldsymbol{n}|$, summarized as follows:

$$\boldsymbol{\theta}^*, \boldsymbol{m}_t^* = \underset{\boldsymbol{\theta}, \boldsymbol{m}_t}{\text{minimize}} \frac{1}{n_t} \sum_{i=1}^{n_t} \mathcal{L}\left(\mathcal{H}\left(x_t^i; \boldsymbol{\theta}(\boldsymbol{n} \odot \mathbf{m}_t)\right), y_t^i\right) - \mathcal{L}(\mathcal{H}(x_t^i; \boldsymbol{\theta}), y_t^i), y_t^i) \tag{2}$$

$$\text{s.t. } |\boldsymbol{m}_t^*| \ll |\boldsymbol{n}|.$$

### 3.2 Neuron-wise Mask

To obtain an optimal subnetwork from the hypernetwork $\mathcal{H}$ for any new coming task, we are motivated by HAT [44] and inhibitory synapse [49] and devise a neuron-wise differentiable mask mechanism to search a small set of neurons with the objective Eq.(2). Specifically, for each layer in the hypernetwork $\mathcal{H}$, let each layer associated with a learnable embedding $\boldsymbol{e}_t^l$ ($l \in \{1, 2, \cdots, L\}$) before training a coming task $t$, where $L$ is the number of layers. To determine if a neuron in $\boldsymbol{n}$ should be used for task $t$, we adopt the binary mask $\boldsymbol{m}_t$ to produce the architecture of the subnetwork. However, neurons typically have only two states, i.e., *activated* or *not activated*. To this end, we use the following function to bridge the gap between the layer embedding $\boldsymbol{e}_t^l$ and layer mask $\boldsymbol{m}_t^l \in \boldsymbol{m}_t$:

$$\boldsymbol{m}_t^l = \sigma(\gamma \cdot \boldsymbol{e}_t^l), \tag{3}$$

where $\sigma$ is the sigmoid function. $\gamma$ is the scaling factor to control the sharpness of the function that can make the binarized progressively [44, 50]. We note that the optimal mask $\boldsymbol{m}_t^*$ can be received by jointly optimizing the model parameters $\boldsymbol{\theta}$ of task $t$, as the task learning is conditioned on mask $\boldsymbol{m}_t$. For instance, the forward propagation of hypernetwork $\mathcal{H}$ during the training of task $t$ can be expressed as follows:

$$\text{forward: } \boldsymbol{h}_t^l = \boldsymbol{h}_t^l \odot \boldsymbol{m}_t^l, \tag{4}$$

where $\boldsymbol{h}_t^l$ denotes the output of neruons in the layer $l$. Correspondingly, the backward propagation can be summarized as follows,

$$\text{backward: } \theta_{lij} = \theta_{lij} - \frac{\partial \mathcal{L}}{\partial \theta_{lij}} \odot \max(m_t^{l,i}, m_t^{l-1,j}), \tag{5}$$

where $\theta_{lij} \in \boldsymbol{\theta}$ refers to a parameter (i.e., weight) between the neuron $i$ of $l$-th layer and the neuron $j$ of $(l-1)$-th layer. $m_t^{l,i}$ denotes the mask value of $i$-th neuron. $\max(m_t^{l,i}, m_t^{l-1,j})$ indicates that the parameter will not be updated if neither the neuron $i$ of $l$-th layer nor the neuron $j$ of $l-1$-th

layer is activated for task $t$. Otherwise, the parameter will be updated. However, the above backward propagation inevitably results in the catastrophic forgetting issue since the weights trained in the previous tasks could be changed. Hence, we regard the used neurons in the previous tasks as the synapses that only take the role of sending messages between different layers. To this end, we devise a cumulative mask to maintain the signal that all previously learned parameters should be frozen. The cumulative mask $\boldsymbol{m}_{\leq t}$ for any task $t$ can be recursively obtained as follows,

$$\boldsymbol{m}_{\leq t} = \max(\boldsymbol{m}_{\leq t-1}, \boldsymbol{m}_t). \tag{6}$$

Note that we set $\boldsymbol{m}_{\leq 0} = \boldsymbol{0}$, indicating all neurons can be used before training task 1. Thus, the Eq.(5) can be revised as follows,

$$\text{backward:} \ \theta_{lij} = \theta_{lij} - \frac{\partial \mathcal{L}}{\partial \theta_{lij}} \odot \max(m_t^{l,i} \odot (1 - m_{<t}^{l,i}), m_t^{l-1,j} \odot (1 - m_{<t}^{l,j})). \tag{7}$$

It is worth noting that mask operation may also pose capacity problems. To this end, we add a regularization term $\mathcal{L}_r$ to the objective $\mathcal{L}$ to hold more room for future tasks as follows,

$$\mathcal{L}_r = \eta \frac{\sum_{l=1}^{L-1} \sum_{i=1}^{n_l} m_t^{l,i} \left( 1 - m_{<t}^{l,i} \right)}{\sum_{l=1}^{L-1} \sum_{i=1}^{n_l} (1 - m_{<t}^{l,i})}, \tag{8}$$

where $n_l$ denotes the neuron number of $l$-th layer in hypernetwork $\mathcal{H}$ and $\eta$ is a hyperparameter that controls the capacity preference for the current task $t$. Note that we provide more details of the hyperparameter setting in Appendix A.

### 3.3 Data-free Replay

To enhance the knowledge transfer from the current task to the earlier tasks, data-free replay recalls network experiences to address the unavailability of old samples or possible data privacy concerns. Recent zero-shot learning solutions [51, 35] provide us with a new perspective. That is, we craft past knowledge through the subnetwork rather than seeking past samples.

**Output Space Modeling.** Let $\boldsymbol{o}_t$ denote the output space (i.e., Softmax space for classification problems) of task $t$ over the subnetwork $\mathcal{H}(\cdot, \boldsymbol{\theta}(\boldsymbol{n} \odot \boldsymbol{m}_t))$. Recent studies [51, 35] reveal that any output representation can be sampled from a Dirichlet distribution as its ingredients fall in the range of [0,1] and their sum is 1. Specifically, the distribution to represent the (Softmax) outputs $\boldsymbol{o}_t^c$ of $c$-th class can be modeled as $Dir(C_t, \beta \times \boldsymbol{\alpha}^c)$, where $c \in \{1, 2, \cdots, C_t\}$ is the class index regarding task $t$, $\boldsymbol{\alpha}^c \in \mathbb{R}^{C_t}$ is the concentrate vector to model $c$-th class, $\beta$ is a scaling parameter [52, 51], and any real value in $\boldsymbol{\alpha}^c$ is greater than 0. It should be noted that there could exist interactive correlations between different classes in a single task since the samples related to different classes may be similar. To mitigate the risk of ignoring the inherent distribution of different classes of samples, we are inspired by [51] and additionally preserve a class similarity matrix $M_t$ after training each task $t$ to describe the correlation between different classes. As $M_t$ of task $t$ is a class-wise table, it will not bring the significant capacity issue. We detail the motivation of the similarity matrix in Appendix B.

**Replay with Impression Craft.** Due to the unavailability of old task samples, like the human memory mechanism, we attempt to synthesize the input of $\mathcal{H}(\cdot, \boldsymbol{\theta}(\boldsymbol{n} \odot \boldsymbol{m}_t))$, which can be denoted as an impression regarding a raw sample in an earlier task. To be specific, we synthesize an impression set $\mathcal{I}_t$ based on the Dirichlet sampling. For any synthesized sample $\hat{x}_t^{c,i}$ corresponding to $c$-th class in task $t$, we initialize $\hat{x}_t^{c,i}$ to random noise, e.g., by sampling from a uniform distribution. And we optimize it with the following objective over a sampled outputs $\hat{\boldsymbol{o}}_t^c$ and the subnetwork $\mathcal{H}(\cdot, \boldsymbol{\theta}(\boldsymbol{n} \odot \boldsymbol{m}_t))$, i.e.,

$$\hat{x}_t^{c,i} = \underset{x}{\mathrm{argmin}} \mathcal{L}_{IC}(\mathcal{H}(\cdot, \boldsymbol{\theta}(\boldsymbol{n} \odot \boldsymbol{m}_t), \tau), \hat{\boldsymbol{o}}_t^c), \tag{9}$$

where $\tau$ is a temperature value [53] used in the output (Softmax) head. In this manner, we can successively produce impressions for each class. Intuitively, we can produce the impressions of different classes equally. However, the subnetwork may show different performances in different classes. That is, we need to consider the emergence of hard classes. Thus, we generate a biased number of impressions of different classes based on the accuracy performance. For each task learning, we report the error rate of each class and normalize them as the distribution of the sampling rate.

**Algorithm 1:** Incremental learning with DSN.

---

**Input:** $\{D_t\}_{t=1}^T$, a hypernetwork $\mathcal{H}$ with neurons $\boldsymbol{n}$ and associated parameters $\boldsymbol{\theta}$, a cumulative mask $\boldsymbol{m}_{\leq 0}$.

Randomly initialize $\boldsymbol{\theta}$ and set $\boldsymbol{m}_{\leq 0} = \boldsymbol{0}$;

**for** $t = 1, 2, \ldots, T$ **do**

    Initialize layer embeddings $\boldsymbol{e}_t = \{\boldsymbol{e}_t^l\}_{l=1}^L$;

    Use $\boldsymbol{e}_t = \{\boldsymbol{e}_t^l\}_{l=1}^L$ to obtain the mask $\boldsymbol{m}_t$ according to Eq.(3);

    Train the hypernetwork $\mathcal{H}$ with $\boldsymbol{\theta}(\boldsymbol{n} \odot \boldsymbol{m}_t)$ on $D_t$ according to Eq.(2) and Eq.(8);

    Obtain the optimal layer embeddings $\boldsymbol{e}_t$ and layer masks $\boldsymbol{m}_t^*$;

    Update the cumulative mask $\boldsymbol{m}_{\leq t}$ via Eq.(6);

    **if** $t > 1$ **then**

        Measure the task similarity scores $S_t$ via Eq.(10);

        Obtain the most similar task $\text{argmax}(S_t)$ with number of class $C_{\text{argmax}(S_t)}$;

        Compute the replay number of each class $\{B_i\}_1^{C_{\text{argmax}(S_t)}}$ according to the class-specific error rate;

        Initialize impression set $\mathcal{I}_{\text{argmax}(S_t)} \leftarrow \{\}$;

        **for** $c = 1 : C_{argmax(S_t)}$ **do**

            Set the concentration parameter $\boldsymbol{\alpha}^c = M_{\text{argmax}(S_t)}^c$;

            **for** $b = B_1, B_2, \cdots, B_{C_{argmax(S_t)}}$ **do**

                **for** $i = 1 : b$ **do**

                    Sample $\hat{o}_{\text{argmax}(S_t)}^c \sim Dir(C_{\text{argmax}(S_t)}, \beta_b \times \boldsymbol{\alpha}^c)$;

                    Initialize $\hat{x}_{\text{argmax}(S_t)}^{c,i}$ to random noise and craft $\hat{x}_{\text{argmax}(S_t)}^{c,i}$ via Eq.(9);

                    $\mathcal{I}_{\text{argmax}(S_t)} \leftarrow \mathcal{I}_{\text{argmax}(S_t)} \cup \hat{x}_{\text{argmax}(S_t)}^{c,i}$;

        Merge mask for task $\text{argmax}(S_t)$: $\hat{\boldsymbol{m}}_{\text{argmax}(S_t)} = \max(\boldsymbol{m}_{\text{argmax}(S_t)}^*, \boldsymbol{m}_t^*)$;

        Fine-tune the task $\text{argmax}(S_t)$ based on $\mathcal{I}_{\text{argmax}(S_t)}$ and $\hat{\boldsymbol{m}}_{\text{argmax}(S_t)}$;

        **if** *accuracy performance of task argmax($S_t$) is higher than before* **then**

            update the mask $\boldsymbol{m}_{\text{argmax}(S_t)}^*$;

**Output:** Optimal hypernetwork $\mathcal{H}$ with parameters $\boldsymbol{\theta}^*$ and task-specific masks $\{\boldsymbol{m}_t^*\}_1^T$.

---

## 3.4 Knowledge Transfer-enhanced Incremental Learning with DSN

**Incremental Learning** ($t \geq 1$). For the first task, i.e., $t = 1$, we randomly initialize the parameters $\boldsymbol{\theta}$ in the hypernetwork $\mathcal{H}$. For each task $t$ including the first task, we start with randomly initializing a task-specific classifier (e.g., a fully-connected layer with a Softmax function) and the corresponding layer embeddings $\boldsymbol{e}_t$. Then, we attempt to jointly learn the model parameters in the hypernetwork $\mathcal{H}$ and task-specific masks of subnetworks regarding each task, where each task-specific mask determines which neurons will be used in the current task. Specifically, given a task $t$, we optimize the parameter $\boldsymbol{\theta}$ and its mask $\boldsymbol{m}_t$ with the objectives of Eq.(2) and Eq.(8) where the update of parameters will follow the rule of Eq.(7) for the purpose of addressing the CF problem. After the training, we likewise build the class similarity matrix $M_t$ and update the cumulative mask $\boldsymbol{m}_{\leq t}$.

**Knowledge Transfer** ($t > 1$). To consolidate the possibly useful knowledge behind the newly learned task to the earlier tasks, we first measure the task similarity and then employ our date-free replay to transfer the knowledge to the most similar task. Notably, we could choose multiple similar tasks to transfer, but this would take more transferring time than it is worth due to the cost of memory replay. Since the unavailability of any real samples in the earlier tasks, it is impossible to use the naive solution for task similarity measurement, i.e., estimating the similarity between the instance distributions of two tasks. Fortunately, we can use the masks of tasks to measure the task similarities. For a current task $t$, we can compute the similarity scores $S_t$ as follows,

$$S_t = \{cosine(\boldsymbol{m}_t, \boldsymbol{m}_0), cosine(\boldsymbol{m}_t, \boldsymbol{m}_1), \cdots, cosine(\boldsymbol{m}_t, \boldsymbol{m}_t - 1)\}, \tag{10}$$

where $cosine$ is the cosine distance. As such, we can obtain the most similar task that owns the largest similarity score, i.e., $\text{argmax}(S_t)$. After that, we can use the data-free memory replay to produce impression crafts as the input samples of the most similar task by optimizing the objective of Eq.(9). It is noted that we only use the impressions to determine whether we need to adjust the subnetwork architecture of the most similar task. Meanwhile, any parameter update in the hypernetwork will cause interference with the current task as well as all the previous tasks. Instead, we only allow the parameter update in the task-specific classifier (head) while freezing any parameters in the subnetwork.

Specifically, we first merge the optimal mask of current task $t$ to the most similar task $\mathrm{argmax}(S_t)$, i.e., $\hat{\boldsymbol{m}}_{\mathrm{argmax}(S_t)} = \max(\boldsymbol{m}^*_{\mathrm{argmax}(S_t)}, \boldsymbol{m}^*_t)$. Then, we fine-tune the task $\mathrm{argmax}(S_t)$ based on $\hat{\boldsymbol{m}}_{\mathrm{argmax}(S_t)}$ and make knowledge transfer possible. We will update the architecture of the subnetwork regarding task $\mathrm{argmax}(S_t)$ if it performs better. Notably, the cumulative mask $\boldsymbol{m}_{\leq t}$ does not need to change due to the fact of $\hat{\boldsymbol{m}}_{\mathrm{argmax}(S_t)} \subseteq \boldsymbol{m}_{\leq t}$. The complete workflow is summarized in Algorithm 1.

## 4   Experiments

**Datasets.**   We employ four benchmark datasets for TIL problem as follows: Permuted MNIST (**PMNIST**) [3], Rotated MNIST (**RMNIST**) [21], Incremental **CIFAR-100** [54, 55], and **TinyImageNet** [19]. PMNIST encompasses 10 variations of MNIST [56], wherein each task is transformed by a fixed permutation of pixels. RMNIST also comprises ten versions of MNIST, with each task rotated by a specific angle between 0 and 360 degrees. The original CIFAR-100 was divided into 20 tasks, each containing 5 different categories. TinyImageNet constitutes a variant of ImageNet [57], we construct twenty 10-way classification tasks for consistency in our experiments.

**Baselines.** We compare our DSN with the following representative baselines: **SGD** [58] utilizes a feature extractor without any parameter update and a learnable classifier to solve a series of tasks incrementally. **EWC** [3] purposefully uses Fisher information as a simple and effective standard baseline to alleviate CF. **IMM** [25] aims to penalize parameter modifications, yielding two variants, i.e., Mean-IMM and Mode-IMM. **PGN** [9] attempts to expand the task network by incorporating a fixed number of neurons. **DEN** [36] dynamically decides the number of new neurons via selective retraining and network split. **RCL** [21] is a reinforcement learning method that controls the scale of newly added neurons while rendering weights relative to the used neurons unchangeable. **HAT** [44] leverages a masking mechanism to prevent the optimization of old neurons. **SupSup** [22] solves the CF problem when sequentially learning tasks by discovering super masks (sub-networks). **WSN** [19] tries to learn the model parameters sequentially and select the optimal sub-network for each task.

**Implementations and Evaluation Metrics.**   In accordance with the task-incremental continual learning framework, all methods used in the experiments employ a multi-head configuration. For two variants of MNIST, we adopt the experimental setup outlined in [44] in which we use a two-layer MLP, a multi-head classifier, and begin with 2000-2000-10 neurons for the first task. For the other datasets, we refer to previous works [19, 44] and use the modified model with three convolutional layers and two fully-connected layers. We follow previous studies [3, 12, 19] and evaluate all methods on three metrics: **ACC**, **BWT**, and **Trans**. **ACC** is a common CL metric that reports the average accuracy of all tasks validated on their respective test sets after training on all tasks. **BWT** measures how the model performance changes for the old tasks once the model is trained on the new task. **Trans** measures the proficiency to transfer knowledge from previously learned tasks to newer ones, indicating the usefulness of the acquired knowledge in facilitating new learning tasks. Notice that more details of the implementation setup are specified in Appendix C. The source codes are available at https://github.com/shanxiaojun/DSN.

Table 1: Performance comparison of the proposed method and baselines on four datasets.

| Model | PMNIST | | | RMNIST | | | CIFAR-100 | | | TinyImageNet | | |
|---|---|---|---|---|---|---|---|---|---|---|---|---|
| | ACC(%) | BWT(%) | Trans(%) | ACC(%) | BWT(%) | Trans(%) | ACC(%) | BWT(%) | Trans(%) | ACC(%) | BWT(%) | Trans(%) |
| SGD | 81.37 | -24.52 | -17.06 | 72.83 | -25.32 | -25.08 | 59.82 | -24.09 | -24.02 | 30.24 | -19.12 | -19.96 |
| EWC | 94.20 | -0.32 | -4.23 | 94.86 | -0.73 | -3.05 | 67.15 | -8.61 | -16.69 | 40.85 | -5.24 | -9.35 |
| mean-IMM | 80.10 | -1.13 | -18.33 | 88.81 | -0.96 | -9.10 | 56.08 | 0.23 | -27.76 | 30.10 | -3.21 | -20.10 |
| mode-IMM | 93.13 | -4.17 | -5.30 | 89.48 | -7.40 | -8.43 | 61.22 | -21.49 | -22.62 | 32.26 | -19.02 | -17.94 |
| PGN | 91.89 | 0.00 | -6.54 | 90.01 | 0.00 | -7.90 | 53.84 | -14.66 | -30.00 | 24.47 | -12.12 | -25.73 |
| DEN | 91.96 | -0.41 | -6.47 | 91.53 | -0.52 | -6.38 | 59.32 | -1.24 | -12.79 | 33.86 | -1.30 | -3.88 |
| RCL | 92.28 | 0.00 | -6.15 | 93.97 | 0.00 | -3.94 | 61.77 | 0.00 | -22.07 | 38.23 | 0.00 | -11.79 |
| HAT | 97.10 | 0.00 | -1.33 | 97.49 | 0.00 | -0.42 | 71.23 | 0.00 | -12.61 | 44.51 | 0.00 | -5.69 |
| SupSup | 97.02 | 0.00 | -1.41 | 97.15 | 0.00 | -0.73 | 71.44 | 0.00 | -12.40 | 43.22 | 0.00 | -6.98 |
| WSN | 97.16 | 0.00 | -1.27 | 97.32 | 0.00 | -0.59 | 72.84 | 0.00 | -11.00 | 45.96 | 0.00 | -4.24 |
| **DSN** | **98.24** | **0.01** | **-0.19** | **97.73** | **0.02** | **-0.18** | **75.17** | **0.02** | **-8.67** | **46.56** | **0.04** | **-3.64** |

**Overall Performance.** To examine the influence of task mixture, we shuffled the tasks with five different seeds, resulting in five lists with different task orders. The averaged results over five runs are presented in Table 1, where the deviation results are reported in Appendix D. We have the following observations: First, not all CL solutions outperform the naive SGD method, even mean-IMM performs slightly worse on PMNIST than SGD. We consider that this observation may be due to the sequential

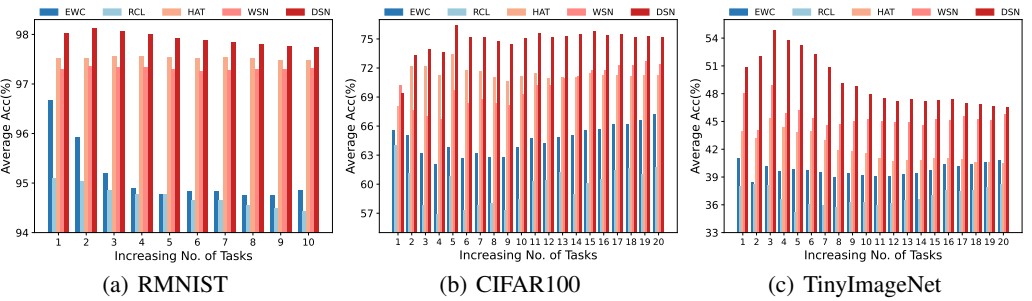

Figure 2: The accuracy performance during entire incremental learning.

weight penalty during knowledge transfer, which could significantly hinder knowledge transfer between old and new tasks. Second, we find that CL methods with network expandability such as PGN and RCL do not perform as well as EWC. This indicates that merely iterative network expansion, without network pruning or neuron selection, has the potential of incurring an over-parameterized risk [13, 59]. On the other hand, HAT, SupSup, and WSN adhere to the merit of network pruning and mainly seek to produce a sub-network by partially selecting some neurons (weights), resulting in higher accuracy performance. However, they can only facilitate forward knowledge transfer, yielding only narrow knowledge consolidation. In contrast, our DSN outperforms all the baselines on all metrics such as achieving the 3.20% improvements on CIFAR-100 regarding ACC. In addition, to show the superiority of DSN in incremental learning from the fine-grained aspect, Fig. 2 presents the averaged results as the number of tasks increases, we can clearly observe that DSN outperforms other baselines during the entire incremental learning process.

**Forward Knowledge Transfer.** The results of Trans in Table 1 demonstrate that DSN is substantially better overall due to the flexible neuron selection. Specifically, DSN performed on PMNIST and RMNIST shows only a slight degradation over the single-task learning scenario. This finding indicates that we can use more compact sub-networks to take on multiple tasks incrementally, which is more practical in future computing contexts, such as edge computing and low resource constraints.

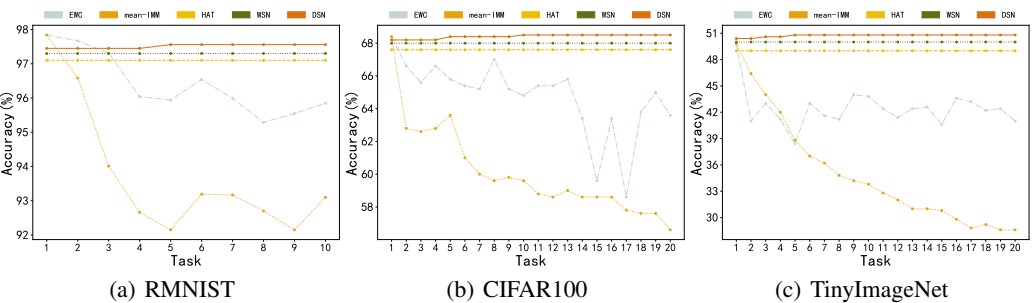

Figure 3: The accuracy performance of the first task in incremental learning.

**Backward Knowledge Transfer.** The experimental results on BWT indicate that SGD suffers from severe knowledge forgetting or negative knowledge transfer issues as it does not take any effort to avoid or alleviate the catastrophic forgetting problem in task incremental learning. Other traditional CL approaches, such as EWC and IMM, choose to confine the shift of model parameters to maintain the old task's performance while learning new tasks sequentially. However, the penalty of parameters still affects the ability of the model to recall old tasks, and these methods also confront severe catastrophic forgetting problems. In comparison with forget-free methods like RCL, HAT, and WSN (BWT=0), DSN can even promote the previous task performance on four datasets (BWT>0). There are two reasons for this observation. First, existing forgetting-free solutions concentrate on simply freezing the neurons or weights used in the previous tasks while making the architecture of the task network regarding each old task unchangeable throughout the entire incremental learning process. As a result, it offers us only barely forgetting or no-forgetting gains. Second, our DSN owning data-free

replay enables the possibility of actual backward transfer, i.e., transferring new knowledge to previous tasks. Fig. 3 shows the performance of the first task throughout the incremental learning. It is obvious that HAT and WSN can only maintain the model performance on the first task while they cannot promote the performance of the first task when incrementally receiving new tasks. This suggests that newly acquired knowledge can help address inadequate experience or knowledge of past tasks.

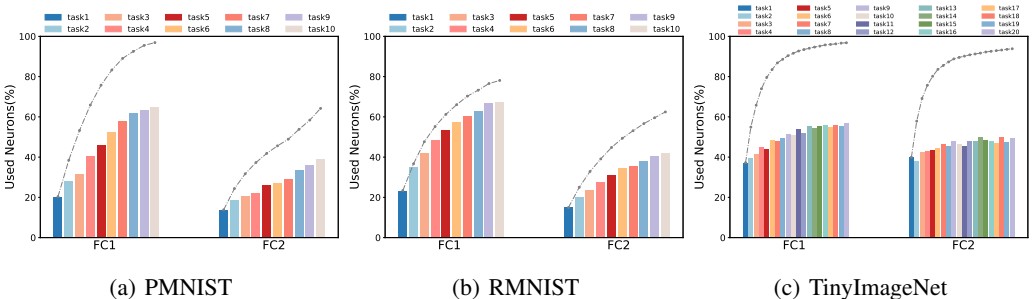

Figure 4: The layer-wise neuron usage in incremental learning.

**Capacity issues.** Fig. 4 shows the capacity monitoring as the sequence of tasks emerges, where we only show the first two fully-connected layers of TinyImageNet for better visualization. From a macro perspective, we can find that the total usage (dashed line) of neurons increases incrementally as more task comes. But we discover that the used neurons for each task are gradually increased, which suggests that DSN prefers to reuse more neurons from earlier tasks when a new task arrives. Especially, DSN reuses more neurons from the past on RMNIST as the total usage increases slowly. The rationale behind this is that although different tasks have different angles of images, they do not change the goal of image classification and those that arrive in sequence are implicitly similar. Furthermore, we investigate the scale impact of our defined hypernetwork. Specifically, we vary the neuron number of each layer in the hypernetwork with different initial learning rates. As shown in Fig. 5, it is evident to observe that a larger over-parameterized hypernetwork enables higher accuracy performance. Interestingly, we find that a smaller learning rate brings us higher gains. We conjecture this phenomenon indicates that over-fitting problem could degrade the model performance.

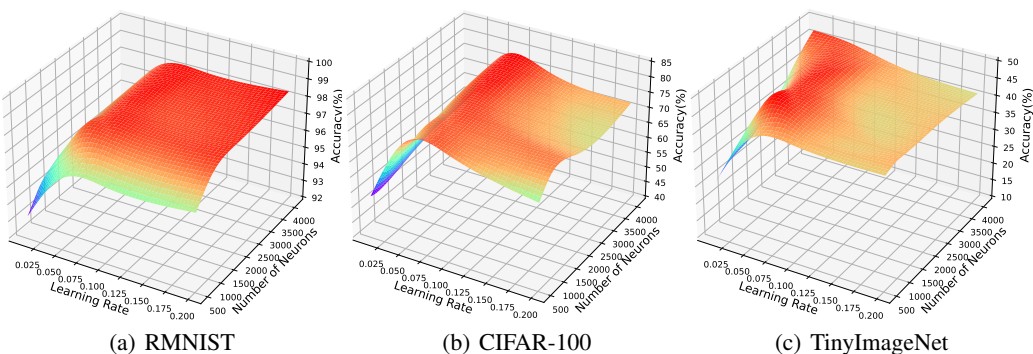

Figure 5: Hypernetwork capacity in incremental learning varying different learning rates.

**Efficiency Issue.** We investigate the efficiency concern regarding our DSN and the representative baselines. First, Table 2 reports the runtime of all approaches including DSN. We first observe that solutions with network expandability take significantly more time, especially for RCL with reinforcement learning. The methods with parameter penalty spend less time while subnetwork selection methods such as WSN cost slightly higher. As for DSN, it costs more time but less than network expandability-based methods. There are two reasons: 1) recent mask mechanisms in recent subnetwork selection methods such as WSN are weight-wise, where the constructed binary masks have the same shape as the weights. As such, we can couple with them in a parallel manner. 2) DSN needs to craft the impressions regarding the old tasks, yielding a higher time cost. Nevertheless, Table 3 shows that DSN and HAT need less trainable masks than WSN due to the neuron-wise masks.

**Sensitivity Analysis.** We evaluate the impact of $\eta$ that aims to hold more room for future tasks. As illustrated in Fig. 6, we can find that a larger $\eta$ will leave more room but bring a slight performance drop. $\eta = 0$ indicates that we do not penalize the number of activated neurons for each task. We can find that the capacity is full while the accuracy performance is the worst. We note that we evaluated the impacts of other key hyperparameters. However, due to space limitations, more details are described in Appendix D.

Table 2: Statistics for full training and inference time, where 'h' represents hours.

| Dataset | SGD | EWC | mean-IMM | mode-IMM | PGN | DEN | RCL | HAT | SupSup | WSN | DSN |
|---|---|---|---|---|---|---|---|---|---|---|---|
| **PMNIST** | 1.24h | 1.36h | 1.42h | 1.38h | 2.52h | 41.54h | 36.18h | 1.58h | 1.55h | 1.65h | 2.43h |
| **RMNIST** | 1.17h | 1.64h | 1.82h | 1.04h | 2.12h | 20.17h | 14.01h | 1.49h | 1.32h | 1.57h | 2.18h |
| **CIFAR-100** | 0.09h | 0.11h | 0.24h | 0.12h | 0.52h | 9.08h | 8.4h | 0.13h | 0.30h | 0.37h | 1.21h |
| **TinyImageNet** | 0.28h | 0.32h | 0.63h | 1.07h | 1.32h | 23.45h | 21.17h | 0.35h | 0.74h | 0.81h | 1.54h |

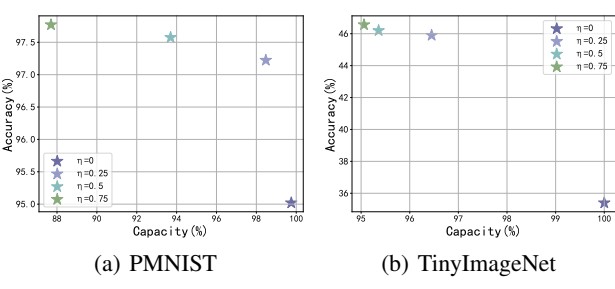

(a) PMNIST  (b) TinyImageNet

Figure 6: The $\eta$ impact on accuracy and network capacity.

Table 3: Total number of masks.

| Dataset | HAT | SupSup |
|---|---|---|
| **P-MNIST** | 4.4K | 40.0M |
| **R-MNIST** | 4.4K | 40.0M |
| **CIFAR-100** | 93.4K | 90.1M |
| **TinyImageNet** | 93.4K | 90.1M |

| Dataset | WSN | DSN |
|---|---|---|
| **P-MNIST** | 40.0M | 4.4K |
| **R-MNIST** | 40.0M | 4.4K |
| **CIFAR-100** | 90.1M | 93.4K |
| **TinyImageNet** | 90.1M | 93.4K |

## 5 Conclusions

Motivated by Lottery Ticket Hypothesis, we introduced a Data-free Subnetworks (DSN) approach for task incremental learning, aiming at enhancing knowledge transfer across sequentially arrived tasks. DSN consists of two components, i.e., neuron-wise mask and data-free replay. The former aims to find an optimal subnetwork using task-specific masks for a new arriving task while the latter attempt to craft the impressions for transferring the knowledge to the past tasks. DSN allows for more flexible knowledge transfer between old and new tasks, whereas backward knowledge transfer was usually ignored before. The experimental results on four datasets show the superiority of our proposed DSN. As part of our future work, we plan to investigate how to devise a knowledge-inspired mask mechanism to enhance knowledge transfer as more tasks arrive sequentially.

## Acknowledgements

This work was supported in part by the National Natural Science Foundation of China (Grant No.62102326 and No.62072077), the Natural Science Foundation of Sichuan Province (Grant No.2023NSFSC1411 and No.2022NSFSC0505), and the Guanghua Talent Project.

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
