## Appendix A: Neuron-wise Mask

### A.1 Details of $\gamma$.

We aim to generate a gated version $\boldsymbol{m}_t^l$ of a single-layer embedding $\boldsymbol{e}_t^l$ to determine the neuron selections. But it is hard to train the layer embedding with backpropagation as demonstrated in [44]. Thus, we follow [44] and apply an annealing strategy on $\gamma$, which is the scaling factor. Specifically, $\gamma$ is annealed during training, inducing a gradient flow and set $\gamma = \gamma_{\max}$. Eq. (3) is to approximate a unit step function as the mask, with $m_t^{l,i} \to \{0, 1\}$ when $\gamma \to \infty$ while $m_t^{l,i} \to 1/2$ when $\gamma \to 0$. In the implementations, we start a training epoch with all units (i.e., neurons) being equally active, which are progressively polarized within the epoch. Consequently, $\gamma$ is annealed as follows,

$$\gamma = \frac{1}{\gamma_{\max}} + (\gamma_{\max} - \frac{1}{\gamma_{\max}})\frac{b' - 1}{\mathcal{T} - 1}, \tag{11}$$

where $\mathcal{T}$ is the total number of batches in an epoch and $b'$ is the batch index. In this manner, we can finally obtain the optimal masks for each task. As depicted in Fig. 1 (a), we can mask the unused neurons and activate task-related neurons.

### A.2 Details of $\eta$.

Intuitively, $\eta \geq 0$ is to control the available capacity for each task. The higher the value of $\eta$, the lower the number of neurons activated in the task. And there is no usage limit if $\eta = 0$. In other words, the setting of $\eta$ can be regarded as a compressibility constant that can affect the compression rate of the learned network for a specific task. A large value of $\eta$ leads to a more sparse network.

## Appendix B: Data-free Mask

### B.1 Details of Dirichlet Distribution.

As we mentioned in the main text, any output representation (i.e., Softmax space) can be sampled from a Dirichlet distribution as its ingredients fall in the range of [0,1] and their sum is 1. Specifically, the distribution to represent the (Softmax) outputs $\boldsymbol{o}_t^c$ of $c$-th class can be modeled as $Dir(C_t, \beta \times \boldsymbol{\alpha}^c)$, where $c \in \{1, 2, \cdots, C_t\}$ is the class index regarding task $t$, $\boldsymbol{\alpha}^c \in \mathbb{R}^{C_t}$ is the concentrate vector to model $c$-th class, $\beta$ is a scaling parameter [52, 51, 35], and any real value in $\boldsymbol{\alpha}^c$ is greater than 0. Intuitively, $\boldsymbol{\alpha}^c$ over class $c$ is to determine how concentrated the probability mass of a sample from a Dirichlet is likely to be. For instance, the mass will be highly concentrated in only a few components, while the rest of the mass will be almost zero. We notice that $\beta$ in $Dir(C_t, \beta \times \boldsymbol{\alpha}^c)$ is the scaling factor that models the spread of the Dirichlet distribution. And the very low values for $\beta$ would yield the highly sparse softmax outputs. As such, our study follows [51] and set $\beta$ in [1, 0.1] for each dataset, which is capable of encouraging higher diversity of softmax outputs.

### B.2 Details of $M_t$.

As we argued that there could exist interactive correlations between different classes in a single task, it is hard to enforce the outputs of a given sample to follow the one-hot representation. Thus, the constructed matrix $M_t$ for task $t$ is to indicate the class similarity. Specifically, we use the task-specific head, e.g., $\boldsymbol{W}_t$ that connects the final and the pre-final layers to generate a normalized class similarity matrix $M_t$. As the head is to specify the final class of a given sample, each neuron in the head corresponds to a class $c$, whereas the affiliated weights of each neuron can be regarded as the template of the class $c$. For instance, each value $M_t^{i,j}$ in $M_t$ reflecting the class similarity between class $i$ and $j$ can be computed by:

$$M_t^{i,j} = \frac{\boldsymbol{W}_t^{i^T}\boldsymbol{W}_t^j}{||\boldsymbol{W}_t^i||\,||\boldsymbol{W}_t^j||}. \tag{12}$$

As each value in 'concentration' parameter $\boldsymbol{\alpha}^c$ over class $c$ is a positive real number, we likewise perform a min-max normalization over each row of the class similarity matrix. As shown in Fig. 7, we can find that the matrix can well show the interactive correlations between different classes in

TinyImageNet dataset. For instance, as shown in Fig. 8, from a visualization point of view, the head weights corresponding to class 'C8' are more similar to class 'C9', which can be clearly caught in the similarity matrix.

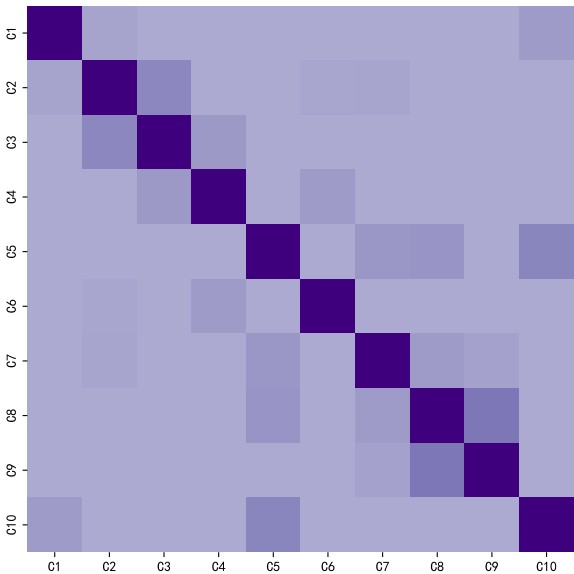

Figure 7: Confusion Matrix of the first task in TinyImageNet.

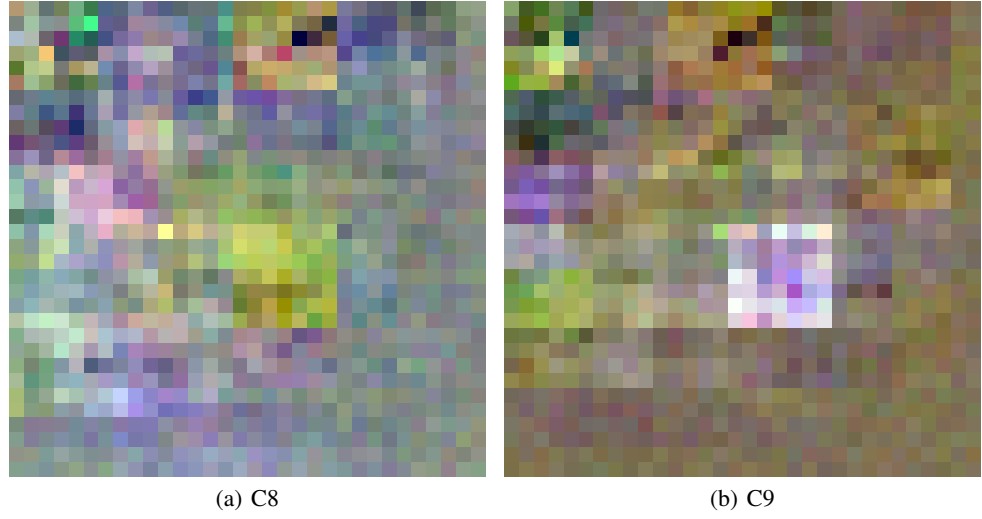

(a) C8           (b) C9

Figure 8: Visualization of head weights in TinyImageNet.

## Appendix C: Experimental Setup

We describe our experimental setup in detail, including the datasets, evaluation protocols, and implementations.

### C.1 Datasets.

The original datasets used are summarized in Table 4. In detail, we conducted the results on four variants, including PMNIST, RMNIST, CIFAR-100, and TinyImageNet. PMNIST and RMNIST

are two variants of the original MNIST dataset [†] containing a large number of $28{\times}28$ monochrome images of handwritten digits. In addition, PMNIST and RMNIST are widely used in incremental learning, where each task of the former is transformed by a fixed and different permutation of pixels, while each task of the latter is rotated by a different angle between 0 to 360 degrees. CIFAR-100 is a CIFAR object recognition dataset with 100 classes. We follow [54] and randomly divide CIFAR-100 into 20 tasks, where each task contains 5 different classes and their examples. TinyImageNet [†] contains 100,000 64×64 colored images in 200 classes. We split twenty 10-way classification tasks from the original TinyImageNet for task incremental learning. For fairness, we randomly sample a subset of the original dataset and also make the testing set the same as the training set following [44]. For PMNIST and RMNIST, we assign each task 60000 images for training and 10000 images for testing to make the task more difficult. For CIFAR-100 and TinyImageNet, we follow the original dataset settings and use 500 train images and 100, 50 testing images for each class respectively.

Table 4: The statistics of four benchmark datasets.

| Dataset | Train | Test | Classes |
|---|---|---|---|
| MNIST [56] | 60,000 | 10,000 | 10 |
| CIFAR-100 [55] | 50,000 | 10,000 | 100 |
| TinyImageNet [19] | 100,000 | 10,000 | 200 |

## C.2 Evaluation Metrics.

To fairly show the model performance and the ability of knowledge transfer (including back and forward knowledge transfer), we use three metrics, i.e., **ACC**, **BWT**, and **Trans**. ACC is a common metric to evaluate the performance of incremental learning. After all tasks are continually well learned, we calculate the average accuracy of all tasks, where the accuracy of each task, denoted by $acc_{T,t}$, is obtained by testing its corresponding test data. To measure backward knowledge transfer, BWT can show the impact of new learning tasks on the accuracy performance of old tasks. Furthermore, BWT$> 0$ indicates the learning of new tasks has a positive impact on old task performance, BWT$< 0$ indicates that the learning of the new task has a negative knowledge transfer on the old task. When BWT is a large negative value, we say that the CL model confronts a *Catastrophic Forgetting* problem. If BWT$= 0$, we say the CL model has no forgetting issue. To measure whether learned knowledge in old tasks facilitates the learning of new tasks, we use Trans to uncover the ability of forward knowledge transfer. Specifically, we treat each task training as a single task learning and compare its accuracy performance with the task performance in the incremental learning context. More formally, these three metrics are defined as follows:

$$\textbf{Average Accuracy: } \mathrm{ACC} = \frac{1}{T} \sum_{t=1}^{T} \mathrm{acc}_{T,t}; \tag{13}$$

$$\textbf{Backward Transfer: } \mathrm{BWT} = \frac{1}{T-1} \sum_{t=1}^{T-1} \mathrm{acc}_{T,t} - \mathrm{acc}_{t,t}; \tag{14}$$

$$\textbf{Forward Transfer: } \mathrm{Trans} = \frac{1}{T} \sum_{t=1}^{T} \mathrm{acc}_{T,t} - \overline{\mathrm{acc}}_t. \tag{15}$$

Herein, $\overline{\mathrm{acc}}_t$ is the task $t$'s accuracy performance in a single task learning manner and $acc_{t,t}$ is the accuracy of task $t$ on the corresponding test data after it is well trained in the CL context.

## C.3 Implementations.

**Hyperparameter Settings.** We implemented our DSN in Python using Pytorch library and all the experiments ran on a single NIVDIA GTX Titan X GPU. We trained all methods, including our DSN, with SGD optimizer. We set the hidden size of GRU to 100; the coefficient of $\eta$ in Eq. (8) is 0.75; the initial learning rate on four datasets is set to 0.05; the temperature value $\tau$ is set to 20; the number

---

[†]`http://yann.lecun.com/exdb/mnist/`
[†]`https://www.kaggle.com/c/tiny-imagenet`

of impressions is set to 1000; and we set $\gamma_{\max}$ to 200. In the Dirichlet distribution, we set $\beta$ in [1, 0.1] for each dataset. To optimize the random noisy image, we employ the Adam optimizer with a learning rate of 0.01, while the maximum number of iterations is set to 1500; and the numbers of training epochs for the task network on PMNIST, RMNIST, CIFAR-100, and TinyImageNet are 30, 30, 50, and 50, respectively.

**Hypernetwork Architecture:** We explain the architecture details of task networks including MLPs and CNNs. Note that our baselines also use the same architectures for a fair comparison.

- *MLP for PMNIST and RMNIST:* We follow [44] and start with 784-2000-2000-10 neurons with RELU activation.

- *CNN for CIFAR-100 and TinyImageNet:* We also follow [44] and extend a modified version of AlexNet for the first task. In more detail, it has three convolutional layers with 64, 128, and 256 filters, with 4×4, 3×3, and 2×2 kernel sizes, respectively, and plus two fully-connected layers of 2048 neurons each. Also, we use rectified linear units as activation and utilize a 2×2 max-pooling operation after three convolutional layers.

**Codes of Baselines:** We have used 10 representative methods for comparison with DSN. SGD is the simplest method and we implemented it by ourselves. For the remaining baselines, we extend their publicly available source codes to conduct the experiments. The url's for the source codes are listed in Table 5.

Table 5: The public source codes of baselines.

| Method | Source |
|---|---|
| EWC [3] | https://github.com/joansj/hat/tree/master/src/approaches |
| IMM variants [25] | https://github.com/joansj/hat/tree/master/src/approaches |
| PGN [9] | https://github.com/joansj/hat/tree/master/src/approaches |
| DEN [36] | https://github.com/jaehong31/DEN |
| RCL [21] | https://github.com/xujinfan/Reinforced-Continual-Learning |
| HAT [44] | https://github.com/joansj/hat |
| SupSup [22] | https://github.com/RAIVNLab/supsup |
| WSN [19] | https://github.com/ihaeyong/WSN |

**Codes of our DSN:** We note that our source codes are submitted as part of the supplementary material. Due to the limit of maximum file size, we provide the full experiment code for CIFAR100 and it can easily be extended to other datasets.

Table 6: The performance deviations on four datasets.

| Model | P-MNIST | | | R-MNIST | | |
|---|---|---|---|---|---|---|
| | ACC (%) | BWT(%) | Trans(%) | ACC(%) | BWT(%) | Trans(%) |
| SGD | 1.178 | 1.825 | 1.178 | 0.623 | 0.533 | 0.623 |
| EWC | 0.705 | 0.068 | 0.705 | 0.244 | 0.087 | 0.244 |
| IMM-mean | 0.860 | 0.117 | 0.860 | 0.620 | 0.215 | 0.620 |
| IMM-mode | 0.546 | 0.510 | 0.546 | 0.342 | 0.235 | 0.342 |
| PGN | 0.264 | 0.000 | 0.264 | 0.126 | 0.000 | 0.126 |
| DEN | 1.258 | 1.258 | 1.258 | 0.241 | 0.176 | 0.241 |
| RCL | 0.410 | 0.000 | 0.410 | 0.335 | 0.000 | 0.335 |
| HAT | 0.584 | 0.000 | 0.584 | 0.191 | 0.000 | 0.191 |
| SupSup | 0.254 | 0.000 | 0.254 | 0.282 | 0.000 | 0.282 |
| WSN | 0.374 | 0.000 | 0.374 | 0.273 | 0.000 | 0.273 |
| DSN | 0.212 | 0.018 | 0.212 | 0.113 | 0.024 | 0.113 |

| Model | CIFAR-100 | | | TinyImageNet | | |
|---|---|---|---|---|---|---|
| | ACC(%) | BWT(%) | Trans(%) | ACC(%) | BWT(%) | |
| SGD | 2.142 | 1.485 | 2.142 | 1.417 | 1.229 | 1.417 |
| EWC | 0.495 | 1.154 | 0.495 | 0.471 | 0.323 | 0.471 |
| IMM-mean | 0.693 | 0.864 | 0.693 | 0.981 | 0.735 | 0.981 |
| IMM-mode | 0.476 | 1.976 | 0.476 | 0.594 | 0.511 | 0.594 |
| PGN | 1.758 | 0.000 | 1.758 | 1.978 | 0.000 | 1.978 |
| DEN | 1.508 | 1.391 | 1.508 | 0.981 | 0.513 | 0.981 |
| RCL | 2.124 | 0.000 | 2.124 | 0.782 | 0.000 | 0.782 |
| HAT | 0.433 | 0.000 | 0.433 | 0.327 | 0.000 | 0.327 |
| SupSup | 0.412 | 0.000 | 0.412 | 0.318 | 0.000 | 0.318 |
| WSN | 0.459 | 0.000 | 0.459 | 0.325 | 0.000 | 0.325 |
| DSN | 0.334 | 0.025 | 0.334 | 0.247 | 0.017 | 0.247 |

## Appendix D: Additional Experimental Results

### D.1 Performance Deviations.

As we have shuffled the tasks with 5 different seeds to alleviate the influence of task mixture in our study, Table 6 reports the standard deviations of model performance on four datasets. We can observe that the achievement of our proposed DSN is stable, which is similar to recent approaches such as SupSup and WSN.

### D.2 Analysis of The Number of Tasks.

In Fig. 9, we show the ultimately used capacity of two fully connected layers (denoted by FC1 and FC2, respectively) under different settings of the number of tasks. Intuitively, as we increase the number of tasks for each incremental learning, the capacity used increases accordingly. For instance, incremental learning that contains 40 tasks uses more capacity than incremental learning that contains 20 tasks. However, in our experiment, we notice that the fact may break the rule occasionally (e.g., handling 50 tasks). A possible reason for this is that DSN may prefer to use previous neurons from learned tasks. On the other hand, DSN could be capable of achieving the model compression as some tasks could be similar. Moreover, as shown in Fig. 10, the extensive experiments show that DSN can always achieve better performance than WSN and HAT as more tasks arrive.

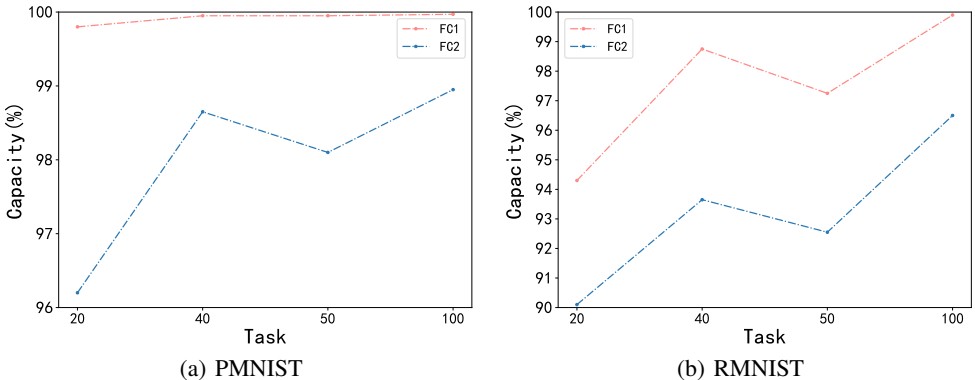

|                 |                 |
| :-------------: | :-------------: |
|   (a) PMNIST    |   (b) RMNIST    |

Figure 9: The finally used capacity when handling different number of tasks.

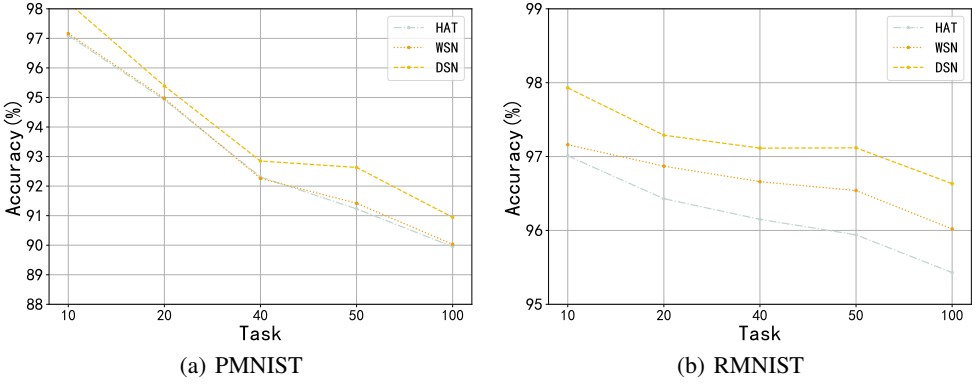

|                 |                 |
| :-------------: | :-------------: |
|   (a) PMNIST    |   (b) RMNIST    |

Figure 10: The average performance under different task numbers.

As shown in Fig. 11, We herein perform incremental learning containing 100 tasks and visualize the change in capacity usage during task learning. As the number of tasks increases, we can clearly find that the capacity used grows rapidly when the task is learned incrementally early, and it starts to slow

down as the number becomes larger. The more tasks we learn, the more knowledge the model gains, and therefore, the easier it is for the model to use the previous knowledge to handle the newly arrived tasks.

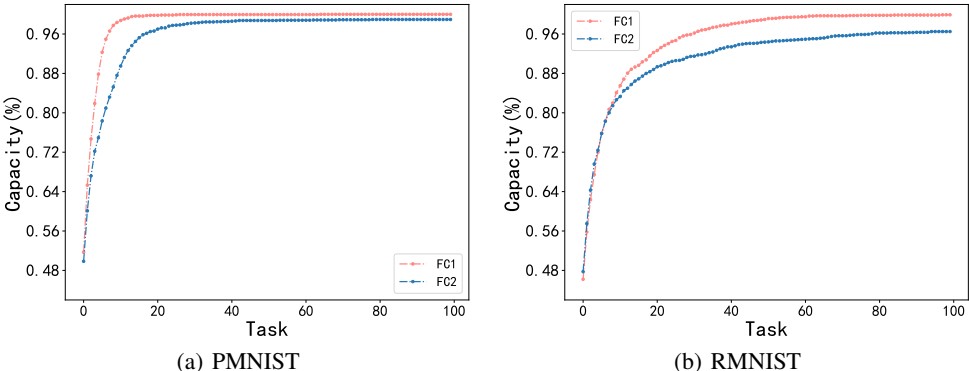

(a) PMNIST        (b) RMNIST

Figure 11: The evolution of the used capacity as the number of tasks increases.

### D.3 Analysis of The Number of Impressions.

As shown in Fig. 12, when the number of impressions escalates within a suitable threshold, the details corresponding to a class are more preserved. However, when the number of impressions is not controlled, the model suffers from overfitting, resulting in a slight degradation of the overall performance. We note that the number of impressions will significantly affect both time resources and restoration resources, whereas it is essential to make a trade-off between cost and gain.

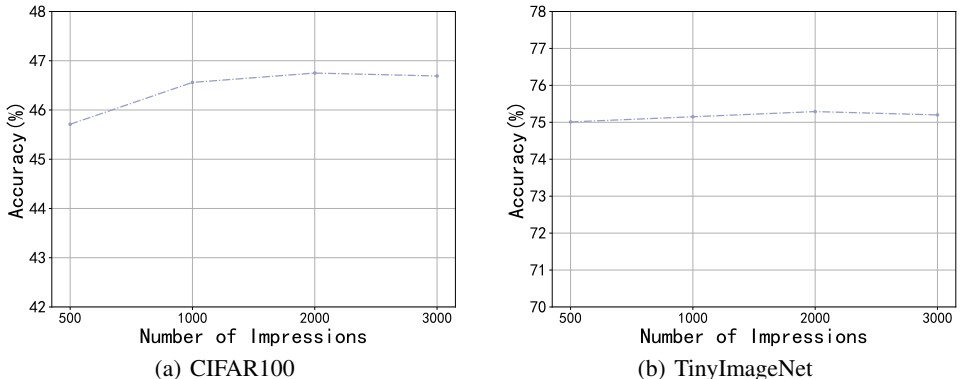

(a) CIFAR100        (b) TinyImageNet

Figure 12: The effect of the number of impressions.

### D.4 Case Study: The Visualization of Impressions.

As shown in Fig. 13, we randomly sample nine images regarding three different classes, and those are generated by our DSN, respectively. We all know that a neural network recognizes the class of an input (e.g., an image) by its latent features (in our case, they are called impressions). The inputs from different classes show different representative features while the samples with the same class are similar, which further indicates that these features/impressions can help our model recall the past learned knowledge regarding different classes. While replaying the samples either sampling directly from the original datasets or employing additional generative methods will result in many costs such as maintaining large generative networks or incrementally picking the samples from the past tasks.

Moreover, maintaining a large and incremental replay buffer will bring data privacy concerns or huge memory costs, which violate the principle of real-world incremental learning context.

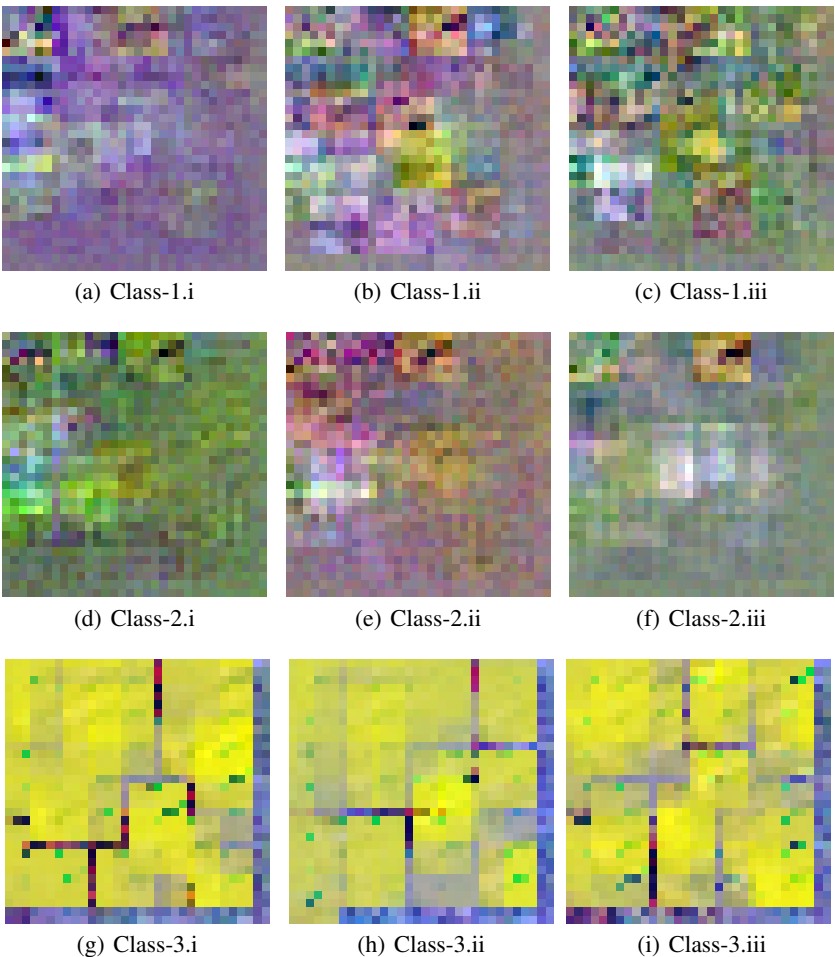

(a) Class-1.i          (b) Class-1.ii          (c) Class-1.iii

(d) Class-2.i          (e) Class-2.ii          (f) Class-2.iii

(g) Class-3.i          (h) Class-3.ii          (i) Class-3.iii

Figure 13: The generated impressions regarding three different classes in TinyImageNet.

### D.5 Similarity Measurement.

In DSN, we use a neuron-wise mask to determine the subnetwork architecture of each arrived task. Besides, due to the unavailability of old samples, we cannot measure task similarity using the approach of [37], which relies on the distribution of data from previous tasks. In our solution, we treat the subnetwork determined by the mask mechanism as knowledge rather than data. Since the subnetworks all originate from the same hypernetwork, with the optimization of Eq.(8), we enforce the new task to reuse more old neurons. Therefore, we measure whether two tasks are similar or not by mask similarity, which is actually subnetwork structure similarity. As each mask corresponds to the neuron in the task network, our DSN that operates mask measurement is vector-level while the weight-based measurement is tensor-level. In a highly dimensional context, in addition to the network scale, there is no doubt that weight-based similarity measurements and even similarity measurements based on task data are affected by the input dimensions, while the mask measurement complexity is only related to the task network scale.

We conduct experiments to examine the effect of our similarity measurement as shown in Table 7. By following [47], we obtain the task similarity between two tasks using their real samples. We obtain the mask similarity in our DSN. Fig. 1 shows the similarity results on PMNIST after training task 10. We can observe that our mask similarity has a similar function to task similarity in [47]. However, as we claimed before, we cannot use the task similarity directly due to the unavailability of old samples.

Table 7: Similarity measurement using different mask mechanisms.

| Arrived Task | Mask Similarity(%) | Task Similarity(%) |
|---|---|---|
| Task 1 | 46.39 | 45.92 |
| Task 2 | 53.26 | 49.32 |
| Task 3 | 57.49 | 54.43 |
| Task 4 | 63.94 | 59.68 |
| Task 5 | 69.42 | 66.05 |
| Task 6 | 72.06 | 75.69 |
| Task 7 | 73.41 | 72.21 |
| Task 8 | 74.75 | 78.79 |
| Task 9 | 78.39 | 80.15 |

## D.6 Transfer Knowledge to Multiple Old Tasks.

We provided experiments that DSN can transfer to more old tasks. According to Table 8, we can observe that transferring new knowledge to multiple old tasks can result in significant time costs (as well as memory costs) that outweigh the benefits. Thus, we only make backward knowledge transfer to the most similar task.

Table 8: The performance results on DNS for knowledge transfer to multiple old tasks.

| Dataset | Multiple Old Tasks | Accuracy(%) | BWT(%) | Elapsed Time |
|---|---|---|---|---|
| PMNIST | No | 98.24 | 0.01 | 2.43h |
| | Yes | 98.28 | 0.03 | 4.75h |
| RMNIST | No | 97.73 | 0.01 | 2.18h |
| | Yes | 97.88 | 0.01 | 3.95h |
| CIFAR100 | No | 75.17 | 0.01 | 1.21h |
| | Yes | 75.34 | 0.01 | 10.92h |
| TinyImageNet | No | 46.56 | 0.01 | 1.54h |
| | Yes | 46.61 | 0.01 | 8.78h |

## D.7 Ablation Study.

In our forward knowledge transfer, the role of the neuron-wise mask is to select an optimal subnetwork architecture for the newly coming task. That is to say, it will affect the new task learning. As for data-free replay, it aims to produce impression crafts for backward knowledge transfer. Hence, this component will significantly affect the accuracy improvements of old tasks. In other words, it is directly related to backward knowledge transfer. We provided the ablation study in Table 9.

Table 9: An ablation study of DSN.

| Dataset | Neuron-wise Mask | Data-free replay | Accuracy(%) | |
|---|---|---|---|---|
| PMNIST | No | No | 97.81 | |
| | No | Yes | 97.99 | |
| | Yes | No | 98.13 | |
| | Yes | Yes | 98.24 | DSN |
| RMNIST | No | No | 97.26 | |
| | No | Yes | 97.42 | |
| | Yes | No | 97.65 | |
| | Yes | Yes | 97.73 | DSN |
| CIFAR100 | No | No | 73.95 | |
| | No | Yes | 74.28 | |
| | Yes | No | 74.81 | |
| | Yes | Yes | 75.17 | DSN |
| TinyImageNet | No | No | 45.90 | |
| | No | Yes | 46.02 | |
| | Yes | No | 46.41 | |
| | Yes | Yes | 46.56 | DSN |

For the neuron-wise mask in the above table, if the choice is "NO", we use the weight-wise mask instead. For Data-free replay, if the choice is "No", we block this module. The above results

demonstrate that removing the neuron-wise mask is more sensitive to the model performance, which also suggests that our neuron-wise mask is better than the weight-wise mask. Moreover, we find that removing the data-free replay component also degrades the model performance, which demonstrates that DSN enables knowledge transfer to the old tasks.

**D.8 Capacity Analysis.**

As Fig. 4 and Fig. 11 show, with the number of tasks increasing, the network does not reach its limit, indicating that DSN effectively utilizes acquired knowledge to handle new tasks. Additionally, we conduct additional experiments to evaluate whether a progressive network can improve accuracy. We enable a random expansion of neurons when a new task arrives (i.e., the state of 'fixed' in the following table is 'No'). The results indicate that the model does not significantly benefit from expansion. Besides, the expansion will lead to a substantial increase in parameters, as demonstrated in Table 10.

Table 10: Comparison results regarding expansion capacity.

| Dataset | Fixed | Accuracy(%) | Layer1 | Layer2 | Layer3 |
|---------|-------|-------------|--------|--------|--------|
| PMNIST | Yes | 98.24 | 2000 | 2000 | N/A |
| | No | 98.29 | 2273 | 2331 | N/A |
| RMNIST | Yes | 97.73 | 2000 | 2000 | N/A |
| | No | 97.75 | 2256 | 2385 | N/A |
| CIFAR100 | Yes | 75.17 | 64 | 128 | 126 |
| | No | 75.21 | 354 | 440 | 509 |
| TinyImageNet | Yes | 46.56 | 64 | 128 | 256 |
| | No | 46.58 | 391 | 345 | 576 |