# OpenReview forum: "Enhancing Knowledge Transfer for Task Incremental Learning with Data-free Subnetwork"
_NeurIPS.cc/2023/Conference — NeurIPS 2023 poster_

### Official Review · Reviewer_Um41 · 2023-06-27

**Soundness:** 2 fair
**Presentation:** 2 fair
**Contribution:** 2 fair
**Rating:** 4
**Confidence:** 4

**Summary:**

This paper presents a Data-free Subnetworks (DSN) approach for task incremental learning. With a random initialized neural network, DSN learns a task-specific neuron-wise mask to find an optimal subnetwork for a new arriving task, and performs data-free replay for transferring the knowledge to the past tasks. DSN achieves superior performance on four benchmarks.

**Strengths:**

- The paper's approach of designing a task-incremental learning method motivated by the Lottery Ticket Hypothesis is reasonable and novel.
- The emphasis on backward knowledge transfer, often overlooked in previous incremental learning methods, is commendable.
- The proposed method outperforms many State-Of-The-Art methods across four benchmarks under multiple evaluation metrics, demonstrating its effectiveness.

**Weaknesses:**

- It is confusing how backward knowledge transfer is achieved through data-free replay. By definition, backward knowledge transfer refers to the process in which new knowledge is transferred to old tasks to potentially improve their performance. However, it seems that the replayed samples (impression crafts) only represent knowledge of old tasks and may not incorporate new knowledge for improving performance in those tasks.

&nbsp;
- The related work section can be further improved by providing detailed comparisons with 1) architecture-based methods (such as [39][41]) that also learn masks to designate different subnetworks for different tasks; 2) incremental learning methods [40][1*] that also consider backward knowledge transfer; and 3) [2*] that is also motivated by Lottery Ticket Hypothesis.
Ref.
[1*] Class-incremental learning with cross-space clustering and controlled transfer, ECCV 2022;
[2*] On the Soft-subnetwork for few-shot class incremental learning, ICLR 2023

&nbsp;
- The method details and formulas need clarification. For example, Eq. (9) lacks an explanation of how to calculate L_{IC}, the optimization objective does not include the variable x, and the discussion on the validity of the Dirichlet distributions representing the task output is less convincing.

&nbsp;
- Knowledge transfer is only performed between current task and the most similar previous task. Considering more similar tasks and evaluating the associated cost could potentially lead to improvements.

&nbsp;
- Some minors: 1) Case errors, such as “Without loss of generality, A classical TIL scenario, ...”; 2) Brackets missing on line 185; 3) Fig. 6 is not referred to in the manuscript.

&nbsp;

**Questions:**

- Which direction(s) does the proposed method belong to, regularization-based, rehearsal-based, or architecture-based?
- What does the f represent in Eq. (1) and (2)?
- What is the difference between a neuron-wise mask and a weight-wise mask?
- What does “bridge the gap between the layer embedding and layer mask” mean?
- Could you provide more details about the structure of the over-parameterized deep neural network used in this method, such as the number of layers and neurons in each layer?
- In Algorithm 1, what do B1, ..., B_{argmax(S_t)} represent?

**Limitations:**

Yes. The authors have discussed the limitations regarding the efficiency and capacity issue of the proposed method.

---

> ### Author Rebuttal · Authors · 2023-08-09
>
> ## 1. Response to "*backward knowledge transfer*"
>
> Thanks for this concern. After training a new task and task similarity measurement, our data-free replay is to produce impression crafts of the most similar task. For backward knowledge transfer, as we claimed that we treated the subnetwork as the knowledge instead of samples, we thus merge the optimal mask of the current task to the most similar task (cf. Knowledge Transfer in Sec. 3.4). This suggests that old task is able to share neurons used in new tasks, enabling the transfer of new knowledge. we then use the impression crafts to update the mask of the old task. This process ensures that the old task's mask is adapted to accommodate the new knowledge, effectively facilitating the transfer of valuable information from the new task to the past tasks. Thus, impression crafts serve as a crucial bridge for carrying over and integrating the latest knowledge into the existing model, leading to improved performance and informed adaptations across the entire task sequence.
>
> ## 2. Response to "*related work*"
>
> Thanks for this suggestion. [1*] proposes two distillation-based objectives for class incremental learning. This method can utilize the feature space structure of the previous model to preserve the representation of previous classes. Besides, a controlled transfer is introduced to approximate and condition the current model on the semantic similarities between incoming and prior classes, maximizing positive forward transfer and minimizing negative backward transfer. However, it can not achieve a positive backward transfer. [40] analyzes the conditions under which updating the learned model of old tasks can be beneficial and lead to backward knowledge transfer. It should be noted that this method is based on SVD, which leads to high computational costs for large dimensional data. [2*] is similar to WSN [3]. Nevertheless, it focuses on a few-shot scene, which is accomplished by jointly learning model weights and adaptive non-binary soft masks, with the major subnetwork minimizing catastrophic forgetting and the minor subnetwork preventing overfitting. Both [2*] and [39] fail to take backward knowledge transfer into consideration. The major contribution of [41] is the induction of the biological neural system into the "M-P neuron model", which sheds light on utilizing the neural network to handle complex tasks.
>
> ## 3. Response to "*Eq.(9)*"
>
> Thanks for this concern. $\mathcal{L}_{IC}$ is the common cross-entropy loss.
>
> ## 4. Response to "knowledge transfer"
>
> Thanks for this concern. The most important contribution of DSN is to enable knowledge transfer from the current task to the previous task (i.e., backward knowledge transfer). The main additional computation costs happen in this procedure due to the impression crafting and the most similar task fine-tuning. In practice, the complexity of the backward knowledge transfer has been considered during the model design investigation. Specifically, transferring newly learned knowledge to multiple similar old tasks can result in significant time costs (as well as memory costs) that outweigh the benefits. Thus, as we claimed in the paper, we only make backward knowledge transfer to the most similar task. According to Table 2, we can find that our proposed DSN does not cause a huge time consumption. Also, we provided the additional experiments that the DSN can transfer to more old tasks **(cf. the Response 1 of #Reviewer bsXa**). We can find that this version can further promote the model's performance. However, the time cost is significantly increased.
>
> ## 5. Response to "*some minors*"
>
> Thanks for pointing out these minors, we will revise them in the next release.
>
> ## 6. Response to "*directions*"
>
> Strictly speaking, our approach DSN falls under architecture-based. Unlike rehearsal-based solutions, we do not retain any samples from the old tasks.
>
> ## 7. Response to "*f*"
>
> Thanks for this concern. $f$ denotes a heypernetwork without any mask.
>
> ## 8. Response to "*weight-wise mask & neuron-wise mask*"
>
> Thanks for this question. The weight-wise mask is one in which each weight in the hypernetwork is associated with a mask value that determines whether or not that weight is used. Note that the size of weight-wise mask equals the number of parameters. In contrast, neuron-wise mask is one in which each neuron in the hypernetwork is associated with a mask value that determines whether or not that neuron is used.
>
> ## 9. Response to "*layer embedding & layer mask*"
>
> Thanks for this question. The layer embedding is the trainable vector. And each mask value is the score within the range of 0 to 1, which numerically determines whether the corresponding neuron should be activated.
>
> ## 10. Response to "*network details*"
>
> Thanks for this concern. We have presented the details of the network architecture in Appendix C.3, as shown below.
>
> - MLP for PMNIST and RMNIST: We follow [39] and start with 784-2000-2000-10 neurons with RELU activation.
>
> - CNN for CIFAR-100 and TinyImageNet: We also follow [39] and extend a modified version of AlexNet for the first task. In more detail, it has three convolutional layers with 64, 128, and 256 filters, with 4×4, 3×3, and 2×2 kernel sizes, respectively, and plus two fully-connected layers of 2048 neurons each. Also, we use rectified linear units as activation and utilize a 2×2 max-pooling operation after three convolutional layers.
>
> ## 11. Response to "Algorithm 1"
>
> Sorry for this confusion, according to the class definition in **Output Space Modeling**, herein, each $B_i$ regarding $C_i$-th of task argmax$(S_t)$ refers to the number of impressions, where $C_{argmax (S_t)}$ indicates that the task argmax$(S_t)$ has $C_{argmax (S_t)}$ classes.

---

> > ### Comment · Reviewer_Um41 · 2023-08-14
> >
> > I appreciate the authors' efforts to respond to all the concerns raised by the reviewers. Clarifications on the concepts and notations are encouraged to be added to the revised manuscript for better understanding. However, I still have some concerns about the role of data-free replay in transferring new task information to past tasks. For me, data-free replay is more likely a way to avoid catastrophic forgetting of past task knowledge after merging a new mask into the old mask, than a way to transfer new knowledge to the old task. Since one major contribution of this paper is positive backward knowledge transfer, I am reluctant to support acceptance at the current stage.

---

> > > ### Author Response · Authors · 2023-08-15
> > > **data-free replay**
> > >
> > > ## Response
> > >
> > > Thanks for the comments and valuable time. We use the impressions obtained by our data-free replay to evaluate whether the merged subnetwork (as we treat the task network as the knowledge) can promote the old task. In our experiments, we obtain positive results (cf. BWT). In practice. if we remove the data-free replay in DSN. DSN also does not confront the CF problem due to the mask mechanism. In fact, catastrophic forgetting and backward knowledge transfer are not equivalent. As opposed to most work on overcoming the problem of catastrophic forgetting, this study centers on an attempt to achieve backward knowledge transfer. However, the migration of new knowledge to old tasks is a tricky endeavor because the principle of continuous learning does not allow access to old samples.
> > >
> > > Existing replay-based methods [1][2] have limitations in achieving forget-free performance and are susceptible to interference from other tasks. Drawing inspiration from the Lottery Ticket Hypothesis, our proposed DSN achieves forget-free performance by employing task-specific masks. Additionally, other architecture-based methods [3][4] also exhibit forget-free, but they neglect the consideration of backward knowledge transfer, and the task-specific masks remain fixed after task training. Our key observation is that while the parameters of neurons used in previous tasks should be protected and can not be updated, there are trainable parameters that continuously adapt with the introduction of new tasks. To illustrate this, let's consider $task_0$. In our DSN approach, optimal task-specific masks are assigned to $task_0$ during training. However, once $task_1$ is trained, the model parameters are updated, and there may be an opportunity to further optimize the architecture for $task_0$. Therefore, we retrain $task_0$ while keeping the parameters of neurons used in both $task_0$ and $task_1$ fixed, ensuring protection against catastrophic forgetting while allowing the model to assign additional neurons specifically for $task_0$, leveraging newly acquired knowledge. Previous methods have also attempted similar approaches [5][6]; however, they preserve previous data as a replay buffer for retraining old tasks, which raises concerns regarding data privacy and memory overhead. To address this, we introduce our data-free replay module, which generates impression crafts that approximate the representation of past tasks [2][7]. If there are any further questions, please do not hesitate to ask. We are delighted to provide a demonstration.
> > >
> > > [1] Tiwari, Rishabh, et al. Gcr: Gradient coreset based replay buffer selection for continual learning. In CVPR, 2022.
> > >
> > > [2] PourKeshavarzi, Mozhgan, et al. Looking back on learned experiences for class/task incremental learning. In ICLR. 2021.
> > >
> > > [3] Kang, Haeyong, et al. Forget-free Continual Learning with Winning Subnetworks, In ICML, 2022.
> > >
> > > [4] Serra, Joan, et al. Overcoming catastrophic forgetting with hard attention to the task. In ICML, 2018.
> > >
> > > [5] Ke, Zixuan, et al. Achieving forgetting prevention and knowledge transfer in continual learning. In NeurIPS, 2021.
> > >
> > > [6] Ke, Zixuan, et al. Continual learning of a mixed sequence of similar and dissimilar tasks. In NeurIPS, 2020.
> > >
> > > [7] Nayak, Gaurav Kumar, et al. Zero-shot knowledge distillation in deep networks. In ICML, 2019.

---

> > > > ### Comment · Reviewer_Um41 · 2023-08-16
> > > >
> > > > Thank you for the comprehensive explanation regarding the data-free replay and the CF problem. I seem to have a slightly distinct perspective on the CF problem. The authors consider the task subnetwork as the knowledge, assuming that retraining the structure and the parameters subnetwork can retain all the task knowledge. However, I tend to view a decline in task performance as indicative of knowledge forgetting. This contradiction might stem from the fact that updating the task-specific classifier is necessary for avoiding performance decline after assigning additional neurons for the task subnetwork. The role of data-free replay, in this context, could be adapting the task-specific classifier to restore the performance of the modified subnetwork on the old task, instead of transferring new task information to old tasks.

---

> > > > > ### Author Response · Authors · 2023-08-17
> > > > > **Data-free replay and CF problem**
> > > > >
> > > > > Thank you for your comments again!
> > > > >
> > > > > **CF:** As noted in the previous comments, sovling the frequently happened CF problem in continual learning scenarios is to avoid or alleviate the accuracy degradation of old tasks after training a new task. For instance, task $t$ obtains the optimal accuracy $A_t$ from the current continual learning system. For any continual learning system defined by previous works, when a new task $t+1$ arrives, the task $t+1$ will be trained on the system. After that, for experimental validation of CF, the task $t$ will be tested again and obtain the new accuracy $A'_t$. If $A'_t<< A_t$, we call the the continual learning system face the CF problem. That is to say, $BWT<0$, where BWT is commonly employed to estimate how the model's performance on previous tasks changes after being trained on new tasks.
> > > > >
> > > > > **backward knoweldge transfer:** In practice, most exsiting continual learning methods take efforts to ensure $BWT \approx 0$ or $BWT=0$ such as WSN and promote the knowledge transfer from the old tasks to the new task. However, how to make $BWT> 0$ is an issue that is currently ignored by the vast majority of studies. We are happy the other reviewers discover this main goal of our work. This is also the the core work of backward knowledge transfer.
> > > > >
> > > > > **data-free replay:** In our data-free replay module, neurons are not provided additionally. In fact, this is because for each task has an equal right to choose any neuron from the hypernetwork that is relevant to it. And each task has already made optimal choices in the training in order to achieve the best accuracy performance. When the new task is arrived and trained, the new learned knowledge provides the old task with the chance to choose again, and if further improvement is gained by the data-free replay operation, it means that the old task benefits from the knowledge gained by the new task. Therefore, we believe this is not a "restore" but a "enhancement", benefited from new knowledge. Because the old task has already obtained the optimal result in the past given the specific hypernetwork, the arrival of the new knowledge makes the optimal result further improve instead of decreasing, which is the benefit of backward knowledge transfer.
> > > > >
> > > > > **our goal:** In practice, we can solve the CF problem even if we only use nueron-wise mask operations in DSN, just like previous studies such as HAT and WSN. In our experiments, by employing data-free replay, we observe positive BWT values while all baselines obtain $BWT \leq 0$. This is the goal of our research, although it is a tricky problem.
> > > > >
> > > > > Therefore, we believe the above explanation responses the concern about new knowledge transfer.

---

### Official Review · Reviewer_cKSd · 2023-07-04

**Soundness:** 4 excellent
**Presentation:** 4 excellent
**Contribution:** 3 good
**Rating:** 6
**Confidence:** 4

**Summary:**

In this work, the authors explore task-incremental learning through the lens of the Lottery Ticket Hypothesis (LTH). They contend, primarily from an LTH perspective, that a distinct task necessitates merely a sparse collection of neurons, hence using only a compact sub-network for its operation. Subsequently, every new task can integrate itself within these compact sub-networks, thereby enabling the overall network to learn new tasks sequentially.

The methodology employed by the authors is intriguing. Initially, a hypernetwork is set up at random, followed by the gradual learning of the model's parameters in tandem with masks that are specific to each task. These masks play a crucial role, determining which neurons and corresponding weights should be utilized or rendered static for the impending task.

Notably, the authors strive to enhance previously acquired knowledge via a process they term 'backward knowledge transfer'. This involves determining the most similar task previously encountered, creating a data impression of that task, and subsequently refining the mask/weights of the preceding tasks. This concept stands as a key contribution of this study.

**Strengths:**


1. The organization of the paper is commendable, effectively guiding readers through the progression of ideas, hypotheses, methodology, results, and conclusions. This well-structured approach allows for a clear understanding of the study and its outcomes.
2. The concept of data-free backward transfer presented in this work is notably intriguing. This aspect, overlooked by existing methods, has been adeptly integrated into the authors' framework, contributing to its uniqueness and potential impact in the field.


3. The authors have executed a compelling array of experiments in their work. Their holistic analysis of their method in relation to Forward Knowledge Transfer, Backward Knowledge Transfer, capacity issues, efficiency, and sensitivity analysis, is thoroughly detailed and insightful. This comprehensive evaluation underscores the robustness of their methodology and enhances the overall credibility of their findings.


**Weaknesses:**

1. The present work bears a significant similarity to the research conducted by Kang et al. [1], which also advocates for a similar framework leveraging the Lottery Ticket Hypothesis (LTH) and employs binary masks to learn task-oriented sub-networks. It would be advantageous for the authors to highlight their unique additions, modifications, or contributions vis-à-vis this recent analogous work, as this would underscore the true value and novelty of the current work.
2. For each new subtask, the presented method learns a mask embedding corresponding to the task. The current manuscript would have been more complete if the authors had discussed the space complexity of the masked embeddings learned with respect to the tasks, comparing it to [1].

Minor:
1. A more concrete review/literature survey of data-free replay should have been performed, For example, somewhat related works like [2] and [3] have not been discussed.  Nonetheless, I encourage authors to explore more related works have utilized data-free replay in various other contexts.

[1] Haeyong Kang, Rusty John Lloyd Mina, Sultan Rizky Hikmawan Madjid, Jaehong Yoon, Mark Hasegawa-Johnson, Sung Ju Hwang, Chang D. Yoo. Forget-free Continual Learning with Winning Subnetworks, In ICML, 2022.

[2] Liu, Huan, Li Gu, Zhixiang Chi, Yang Wang, Yuanhao Yu, Jun Chen, and Jin Tang. "Few-shot class-incremental learning via entropy-regularized data-free replay." In ECCV, 2022.

[3] Choi, Yoojin, Mostafa El-Khamy, and Jungwon Lee. "Dual-teacher class-incremental learning with data-free generative replay." In CVPR Workshops, 2021.


**Questions:**

1. In connection with Objective 8, the authors introduce a capacity constraint to the current task, which may impact the task's performance due to the limitation it imposes on the neural network's capacity. I would be keen to hear an explanation from the authors on the extent to which this constraint influences the efficacy of the current task.

2. Considering the methodology used to measure the accuracy of task $argmax(S_t)$, I notice that the authors operate under the assumption of not having access to the previous task's data. Given this constraint, how do the authors determine the accuracy of the task post fine-tuning $argmax(S_t)$ and subsequently compare it with its preceding accuracy? Could they elaborate on their approach to evaluating accuracy under these specific conditions and how they ensure the validity of their comparisons?



**Limitations:**

In relation to the proposed method, I understand that tasks are stored as parts of a subnetwork. This is an intriguing concept, yet I find myself questioning its scalability when considering a large number of diverse tasks. Given that the total upper capacity of the neural network is fixed, I am uncertain if the proposed method would accommodate scaling up effectively. It seems to me that a broad and varied set of tasks may exceed the inherent capacity limits of the neural network, potentially leading to capacity exhaustion or compromise in task performance.

---

> ### Author Rebuttal · Authors · 2023-08-09
>
> ## 1. Response to "*WSN*"
>
> Thanks for this suggestion.
>
> (1) DSN devises a neuron-wise mask mechanism to select neuron-affiliated weights for new task learning.
>
> (2) DSN enables positive knowledge transfer in both forward and backward directions. In particular, the data-free replay mechanism in DSN regards the trained subnetwork as a past experience and uses it to craft impressions regarding the past samples, which does not require holding any actual samples related to past tasks.
>
> The comparison details:
> (1) WSN mainly relies on the weight-wise mask to select the optimal subnetwork from a hypernetwork. But the selected subnetwork is not a continuous structure of a layer-dependent neural network. Because WSN selects a portion of the weights that are relevant to the neuron. WSN also maintains a binary mask whose size equals the number of parameters, resulting in significant resource consumption. In contrast, DSN devises a neuron-wise mask to choose the optimal subnetwork, which can ensure the completeness of the subnetwork. Meanwhile, since our masks are neuron-level, they are lightweight. According to Table 3, we can find DSN requires less number of masks.
>
> (2) WSN has well addressed the CF problem, but WSN does not incorporate backward knowledge transfer. That is, newly acquired knowledge cannot help facilitate any old task. In contrast, DSN is capable of achieving positive gains on old tasks. Meanwhile, in backward knowledge transfer, we consider the data inaccessibility of the old tasks. And we use the data-free replay to effectively avoid this limitation.
>
> Thus, DSN is significantly different from WSN. We believe that DSN offers a new paradigm that focuses more on positive backward knowledge transfer and consider the concern of data unavailability.
>
> ## 2.Response to "*space complexity*"
>
> For instance, given a task network containing a two-layer MLP and a classifier, each layer consists of n neurons, yielding the length of neuron mask embedding for each layer is $n$. Thus, the space complexity of each is $\mathcal{O}(n)$ due to its size equal to the neuron numbers. Instead, WSN will result in $\mathcal{O}(n^2)$. According to Table 3, we can find that DSN requires less number of masks.
>
> ## 3. Response to "*minor*"
>
> Thanks for your significant suggestions. In the next release, we will thoroughly revise all the typos and also engage in a comprehensive discussion of the related works.
>
> ## 4. Response to "*Objective 8*"
>
> Thanks for this question. In Eq.(8), we set a hyperparameter $\eta$ to control the capacity preference for each new task. The higher the value of $\eta$, the lower the number of neurons activated for the new task. In particular, there is no capacity limit when $\eta =0$.
>
> First, in our main experiment, as shown in Fig. 6, when $\eta =0$, it is evident that all neurons have been utilized. But, the accuracy performance is the worst. Obviously, it is because the model cannot utilize new neurons to learn new knowledge as more new tasks arrive. In addition, a larger $\eta$ indicates that the model will hold more room for new task learning. However, we have found that having an excessively large capacity constraint does not lead to a significant improvement in accuracy performance. In other words, excessive reuse of old neurons can also result in underfitting issues during training for new tasks.
>
> Second, in Appendix D.2, we conducted another experiment where we set up a different number of tasks (up to 100 tasks) to validate the performance, where the hypernetwork settings were the same as in the main experiment. As Fig. 10 shows, compared to HAT and WSN, we first observe that the accuracy of all three methods decreases as the number of tasks increases, mainly due to the fact that the hypernetwork is too small. However, we observe that the DSN always has the best performance as the number of tasks increases.
>
> ## 5. Response to "*old task accuracy*"
>
> Thanks for this question. As we consider the unavailability of old samples regarding task argmax($S_t$), we are motivated by zero-shot learning and treat the trained subnetwork as the past knowledge of task argmax($S_t$). We also have it determined output space. Thus, we can craft the input space of task argmax($S_t$) using data-free replay. As these impression crafts are determined by the network of the old task, we use them to obtain ground-truth accuracy. Then, we merge the optimal mask of the current task to the most similar task. This suggests that the old task is able to share neurons used in new tasks, enabling the transfer of new knowledge. Next, we use impression crafts to update the mask of the old task. DSN decides to maintain the updated mask for the old task only if the updated subnetwork shows a significant improvement in accuracy over the ground-truth.
>
> ## 6. Response to "*capacity limitation*"
>
> Thanks again. As Fig. 4 and Fig. 11 show, with the number of tasks increases, the network does not reach its limit, indicating that DSN effectively utilizes acquired knowledge to handle new tasks. Additionally, we conduct additional experiments to evaluate whether a progressive network can improve accuracy. We enable a random expansion of neurons when a new task arrives. The results indicate that the model does not significantly benefit from expansion. Besides, the expansion will lead to a substantial increase in parameters, as demonstrated below. Your insightful suggestion shad light on a new perspective. We will explore how to effectively utilize the newly added neurons in future work.
>
> | Data | Fixed | Accuracy(%) | Layer1 | Layer2 | Layer3 |
> | --- | --- | --- | --- | --- | --- |
> | PMNIST | Yes | 98.24 | 2000 | 2000 | N/A |
> |     | No  | 98.29 | 2273 | 2331 | N/A |
> | RMNIST | Yes | 97.73 | 2000 | 2000 | N/A |
> |     | No  | 97.75 | 2256 | 2385 | N/A |
> | CIFAR100 | Yes | 75.17 | 64  | 128 | 126 |
> |     | No  | 75.21 | 354 | 440 | 509 |
> | TinyImageNet | Yes | 46.56 | 64  | 128 | 256 |
> |     | No  | 46.58 | 391 | 345 | 576 |

---

> > ### Comment · Reviewer_cKSd · 2023-08-13
> > **Response to the "old task accuracy" still seems vague**
> >
> > Thanks to the authors for providing their response!
> >
> > However, I am still finding it hard to comprehend how the crafted impressions are used to get the "ground-truth accuracy", especially given that these impressions are essentially pseudo-samples. In a similar vein, the authors in their response describe the following, "DSN decides to maintain the updated mask for the old task only if the updated subnetwork shows a significant improvement in accuracy over the ground-truth", again, the process of obtaining the "ground-truth accuracy" remains ambiguous.

---

> > > ### Author Response · Authors · 2023-08-15
> > > **old task accuracy**
> > >
> > > ## Response
> > >
> > > Thank you for your valuable time and comments! After obtaining the optimal subnetwork for the current task $t$, we turn to find its most similar task argmax$S(t)$. We obtain a set of impressions for each class in task argmax$S(t)$ with the optimization of Eq.(9). We then perform knowledge backward transfer by merging the masks of the two tasks and fine-tuning optimization. Before that, in our implementation, we report two indicators as ground truths, i.e., the impression performance on the subnetwork (we call it old subnetwork) of argmax$S(t)$ including accuracy and loss. When fine-tuning the task argmax$S(t)$, the impressions are found to be significantly more accurate on the updated sub-network, or the loss is significantly reduced relative to the old subnetwork, then we decide to update the mask of task argmax$S(t)$.

---

> > > > ### Comment · Reviewer_cKSd · 2023-08-18
> > > > **Thanks for the response.Need more clarity!**
> > > >
> > > > Thanks to the author to provide an elaborate response, I have gained a clearer understanding of the methodology employed for backward transfer.
> > > >
> > > > However, I still have some lingering queries. Specifically, after the masks are merged, can you clarify the dataset on which the fine-tuning is carried out? Is it executed using the class impression, the current task's data, or a combination of both?
> > > >
> > > > Moreover, from your response, I gathered that the ground-truth accuracy is gauged using the generated impression, which originates from the model itself. I'm somewhat tentative about the model's self-generated impression serving as the benchmark for its own performance. It might be beneficial if this aspect could be elucidated further.
> > > >
> > > > Given that backward transfer appears to be a significant contribution of your work, I believe that clarifying these nuances would greatly benefit me and other reviewers.

---

> > > > > ### Author Response · Authors · 2023-08-20
> > > > > **Fine-tuning concern**
> > > > >
> > > > > Thank you again for your comments. We are honored to response insightful concerns raised by you and other reviewers! Here, we clarify a few things about impression setting.
> > > > >
> > > > > **(1) fine-tuning:** In the fine-tuning process, we only use impression crafts. Please allow us to explain the motivation behind this setup. Our DSN provides a clear and independent learning process either in the forward knowledge transfer or backward knowledge transfer. In the forward knowledge transfer process, the new task has obtained the best performance through the hypernetwork and the mask mechanism based on the retained past knowledge. That is to say, its corresponding network structure is deterministically unmodifiable at the present time. When the most similar old task is specified, using both impressions and data from the current task as in multitask learning would cause learning confusion since the fine-tuning process is conditioned on the masking results of the current task. In other words, the parameters of the network structure corresponding to the current task could change again, which would leads to a change of the masks, ultimately interfering with forward knowledge transfer in addition to the time cost.
> > > > >
> > > > > **(2) impression accuracy concern:** In our study, as you commented before, we have an explicit premise that the associated samples will not be accessible after the task training is completed. However, enabling backward knowledge transfer on this premise is extremely challenging but does fit the real purpose of continual learning. To this end, we relying on the crafted impressions by our date-free replay to fine-tune the most similar old task. We detail the imression concern from two aspects.
> > > > >
> > > > > #1 First, for task incremental learning, the goal of an arbitrary task is to learn a model that maximizes the difference in sample representations under different classes. When classes and the model are determined, we can produce impressions with the optimization of Eq. (9). Under the optimization of Eq. (9), the impressions in a way that the subnetwork of the most similar old task strongly believes them to be actual samples that belong to classes. That is say, the role of impressions is to be able to explicitly specify the different classes, rather than focusing on the approximation between the real input samples and the impressions. Based on this principle, we believe the produced impressions can well correspond the their classes.
> > > > >
> > > > > #2 Second, after we have merged the masks for backward knowledge transfer, the initially updated subnetwork structure will not adapt to these impressions (i.e., the updated subnetwork structure will be unable to integrate new and old knowledge based on these impressions, initially.), inevitably leading to training oscillations and *accuracy gaps*. Like the knowledge integration in zero-shot learning, new knowledge needs to be fine-tuned to gradually adapt to the most similar old tasks. In our solution, during the fine-tuning process, the new architecture is determined by continuously training the merged mask, and this training process of the new architecture is to eliminate the accuracy gap mentioned above and further improve the performance accuracy over ground-truth accuracy in order to ensure that the new architecture is reliable.
> > > > >
> > > > > In practice, several measures in our study have been taken to promote the representativeness of these impressions. First, we are motivated by recent zero-shot learning and use the Dirichlet distribution to model the output space, ensuring a differentiated representation of the different class. Second, we consider the interactive correlations between different classes by using a similarity matrix $M_t$，the reason is that some samples from different classes may have some similar characteristics. Third, as we claimed in the main text, the subnetwork regarding the most similar old task may show different performances in different classes. That is, we need to consider the emergence of hard classes. Thus, we generate a biased number of impressions of different classes based on the accuracy performance. For each task learning, we report the error rate of each class and normalize them as the distribution of the sampling rate. In our experiments, as shown in the Appendix D.3 (cf. Fig. 12), we can find our impression crafting is robust in DSN solution.

---

### Official Review · Reviewer_bsXa · 2023-07-06

**Soundness:** 3 good
**Presentation:** 3 good
**Contribution:** 3 good
**Rating:** 6
**Confidence:** 4

**Summary:**

The paper proposes a novel method for task incremental learning that uses a subnetwork for each task. The method is based on neuron masking obtained by learnable embedding. The method also finds the most similar task from the past tasks and allows for backward transfer. The method is tested on four benchmarks: PMNIST, RMNIST, CIFAR-100, and TinyImageNet. The results show an improvement in performance over other methods.

**Strengths:**

-   The proposed method achieves a better performance than other methods on the three studied benchmarks.
-   Up to my knowledge, few papers addressed positive backward transfer in CL. It is nice to see more work addressing this aspect.
-   Analysis of the reusability of the neurons over tasks is provided.
-   Efficiency analysis is considered.
-   Different metrics are evaluated Accuracy, BWT, and Trans, which assess different requirements in CL.


**Weaknesses:**

-  Backward transfer to one only past task is not that convincing. Multiple previous tasks could benefit from the new knowledge if they are similar enough. This is my main concern. Since backward transfer is one of the main contributions, I expect more investigation into that point.
-  Despite the fact that most of the paper is easy to follow, Section 3.3 lacks an overview at the beginning that can link the following information to how it contributes to the method. It becomes more clear after reading Section 3.4.

 [suggestion] Maybe the authors can consider reordering these two sections or clarify a bit more in Sec 3.3
-  The paragraph explaining the backward transfer is very brief (Line 125-221), and some sentences are not clear. More elaboration could be useful. Algorithm 1 is helpful, though.
-  An analysis that assesses the validity of the proposed similarity measure (even on a toy experiment) could be helpful.
-  Minor: text in figures 3 and 4 is not readable.
-  Minor: typo in the last line in Algo.1 “hyep”
-  Some closely related works are missing. Not necessarily to empirically compare with them as you provide a comparison with one recent related method (WSN), but at least to have a discussion in the related work section. Examples of the missing works are below.

o	Gurbuz, Mustafa B., and Constantine Dovrolis. "NISPA: Neuro-Inspired Stability-Plasticity Adaptation for Continual Learning in Sparse Networks." International Conference on Machine Learning. PMLR, 2022.

o	Sokar, Ghada, Decebal Constantin Mocanu, and Mykola Pechenizkiy. "Spacenet: Make free space for continual learning." Neurocomputing 439 (2021): 1-11.

o	Sokar, Ghada, Decebal Constantin Mocanu, and Mykola Pechenizkiy. "Avoiding Forgetting and Allowing Forward Transfer in Continual Learning via Sparse Networks." Joint European Conference on Machine Learning and Knowledge Discovery in Databases. Cham: Springer Nature Switzerland, 2022.

o	Wang, Zifeng, et al. "SparCL: Sparse continual learning on the edge." Advances in Neural Information Processing Systems 35 (2022): 20366-20380.

o	Yin, Haiyan, and Ping Li. "Mitigating forgetting in online continual learning with neuron calibration." Advances in Neural Information Processing Systems 34 (2021): 10260-10272.


**Questions:**

-   What is the contribution of each proposed components (neuron wise mask and data free replay) to the performance gain in DSN? Do you have an ablation for that?
-   L 215 “Instead, we only allow the parameter update in the task-specific classifier (head) while freezing any parameters in the subnetwork.” This sentence is not clear. Does it mean that in backward transfer, you only update the head?
-   Looking at Table 1 and Table 2 together and comparing WSN to DSN, do you think that the improvement gain in performance worths the additional costs (i.e., double the training cost in some cases). In other words, Are there specific cases in that one can definitely choose DSN over WSN?
-   Is the proposed method applicable for more realistic scenarios i.e. class-incremental learning?
-   Do you have some thoughts on how this approach can be further extended/improved to allow for backward transfer to multiple tasks?

**Limitations:**

-	Limitations and social impact are not discussed

---

> ### Author Rebuttal · Authors · 2023-08-09
>
> ## 1. Response to "*multiple previous tasks*"
> Thanks for this concern. We consider that transferring new knowledge to multiple old tasks can result in significant time costs (as well as memory costs) that outweigh the benefits. Thus, we only make backward knowledge transfer to the most similar task. Herein, we also provided experiments that DSN can transfer to more old tasks as below. We find that this version can further promote model performance. However, the time cost is significantly increased.
> | Dataset | Multiple old tasks | Accuracy(%) | BWT(%) | Elapsed Time |
> | --- | --- | --- | --- | --- |
> | PMNIST | No  | 98.24 | 0.01 | 2.43h |
> | PMNIST | Yes | 98.28 | 0.03 | 4.75h |
> | RMNIST | No  | 97.73 | 0.02 | 2.18h |
> | RMNIST | Yes | 97.88 | 0.06 | 3.95h |
> | CIFAR100 | No  | 75.17 | 0.02 | 1.21h |
> | CIFAR100 | Yes | 75.34 | 0.07 | 10.92h |
> | TinyImageNet | No  | 46.56 | 0.04 | 1.54h |
> | TinyImageNet | Yes | 46.61 | 0.06 | 8.78h |
>
> ## 2. Response to "*Sec. 3.3*"
> Thanks for your insightful suggestions. In the upcoming release, we will present an overview of this section. In addition, we will thoroughly revise all the typos and also engage in a comprehensive discussion of the related works.
>
> ## 3. Response to "*Ablation study*"
> Thanks for this question. In our forward knowledge transfer, the role of the neuron-wise mask is to select an optimal subnetwork architecture for the newly coming task. That is to say, it will affect the new task learning. As for data-free replay, it aims to produce impression crafts for backward knowledge transfer. Hence, this component will significantly affect the accuracy improvements of old tasks. In other words, it is directly related to backward knowledge transfer.
> We provided the ablation study as follows.
>
> | Dataset | Neuron-wise mask | Data-free replay | Accuracy(%) |
> | --- | --- | --- | --- |
> | PMNIST | No  | Yes | 97.99 |
> |     | Yes | No  | 98.13 |
> |     | Yes | Yes | 98.24 |
> | RMNIST | No  | Yes | 97.42 |
> |     | Yes | No  | 97.65 |
> |     | Yes | Yes | 97.73 |
> | CIFAR100 | No  | Yes | 74.28 |
> |     | Yes | No  | 74.81 |
> |     | Yes | Yes | 75.17 |
> | TinyImageNet | No  | Yes | 46.02 |
> |     | Yes | No  | 46.41 |
> |     | Yes | Yes | 46.56 |
>
> For the neuron-wise mask in the above table, if the choice is "NO", we use the weight-wise mask instead. For Data-free replay, if the choice is "No", we block this module. The above results demonstrate that removing the neuron-wise mask is more sensitive to the model performance, which also suggests that our neuron-wise mask is better than the weigh-wise mask. Moreover, we find that removing the data-free replay component also degrades the model performance, which demonstrates that DSN enables the knowledge transfer to the old tasks.
>
> ## 4. Response to "*L 215*"
>
> Sorry for the confusion. During backward knowledge transfer, we will not change any parameters in the hypernetwork as it could cause interference with other tasks. But, we update the masks of the most similar old task as well as the task-specific head. That is to say, the subnetwork architecture regarding the most similar old task can be adjusted while keeping the parameters fixed. In this way, this operation will not cause interference with other old tasks as well as the newly coming task.
>
> ## 5. Response to "WSN and DSN"
>
> Thank you for this great question. We have the following explanation.
> WSN mainly relies on the weight-wise mask to select the optimal subnetwork from a hypernetwork. In practice, the selected subnetwork is not a continuous structure of a layer-dependent neural network. This is because the WSN selects a portion of the weights that are relevant to the neuron. This results in the inability to separate the subnetwork selected from the hypernetwork into a single but complete network architecture. That is to say, every task in the continual learning system would need to download this huge hypernetwork in order to complete the model inference. In the future scenarios such as pervasive computing may be unrealistic. In addition, WSN needs to maintain a binary mask whose size equals the number of parameters also results in significant resource consumption.
>
> WSN and many other prior works have well addressed the CF problem, i.e., zero forgetting or forgetting-free, but they do not consider backward knowledge transfer well. That is, newly acquired knowledge cannot help facilitate any old task. In contrast, our DSN is capable of achieving positive gains on old tasks. Meanwhile, in the process of backward knowledge transfer, we also consider the data inaccessibility of the old tasks. And we use the data-free replay mechanism to effectively avoid this limitation.
>
> Therefore, we believe that our DSN offers a new paradigm that focuses more on positive backward knowledge transfer and takes into account the concern of data unavailability for old tasks.
>
> ## 6. Response to "*class-incremental learning*"
>
> Thanks for this question. In practice, our proposed DSN can apply to more contexts including class-incremental learning. [1] also gives a detailed theoretical study that task-incremental learning can be naturally transformed into class-incremental learning with the help of out-of-distribution detection.
> [1]Kim et al. A theoretical study on solving continual learning. NeurIPS 2022.
>
> ## 7. Response to "*future plan*"
> Thanks again, we plan to devise an efficient machinsm that aims to craft the representative impressions. Besides, we conduct backward transfer on past tasks in a sequential order, sorting them based on their similarity to the current task. We monitor the performance of past tasks after backward transfer. If their performance improves, we proceed to the next task in the sequence. We continue this process until the performance of the old tasks no longer shows enhancement from knowledge transfer. This iterative approach ensures that we prioritize and focus on transferring knowledge to the most relevant tasks first.

---

> > ### Comment · Reviewer_bsXa · 2023-08-15
> > **Official Comment by by Reviewer bsXa**
> >
> > Thank you for your response and for providing extra results. I have read your rebuttal.
> >
> > I am quite surprised that transferring the knowledge to multiple previous tasks led to a minor improvement in performance. Can the authors comment on that?
> >
> > For the ablation study, it is nice to have a baseline that both components are not used such that we can assess the effect of each.

---

> > > ### Author Response · Authors · 2023-08-15
> > > **accuracy**
> > >
> > > Thank you for your valuable time and suggestions. First of all, the results in the table above are average accuracy rates. Second, during the backward transfer of multiple old tasks, it is difficult to further improve the accuracy by mask merging and classifier updating due to the decreasing similarity of the old tasks. That is, the old tasks with lower similarity can hardly benefit from knowledge transfer.
> > >
> > > With regard to the second recommendation. We need some time to prepare such a baseline. We will update the comments later (we have updated the table in the following comment. Thanks for your great suggestion!).

---

> > > > ### Author Response · Authors · 2023-08-16
> > > > **a base model in ablation study**
> > > >
> > > > In the following table, we additionally provide a base model that removes two modules from the DSN and uses the weight-wise mask for continual learning.
> > > >
> > > > | Dataset      | Neuron-wise mask | Data-free replay | Accuracy(%) |     |
> > > > | ------------ | ---------------- | ---------------- | ----------- | --- |
> > > > | PMNIST       | No               | No               | 97.81       |     |
> > > > |              | No               | Yes              | 97.99       |     |
> > > > |              | Yes              | No               | 98.13       |     |
> > > > |              | Yes              | Yes              | 98.24       | DSN |
> > > > | RMNIST       | No               | No               | 97.26       |     |
> > > > |              | No               | Yes              | 97.42       |     |
> > > > |              | Yes              | No               | 97.65       |     |
> > > > |              | Yes              | Yes              | 97.73       | DSN |
> > > > | CIFAR100     | No               | No               | 73.95       |     |
> > > > |              | No               | Yes              | 74.28       |     |
> > > > |              | Yes              | No               | 74.81       |     |
> > > > |              | Yes              | Yes              | 75.17       | DSN |
> > > > | TinyImageNet | No               | No               | 45.90       |     |
> > > > |              | No               | Yes              | 46.02       |     |
> > > > |              | Yes              | No               | 46.41       |     |
> > > > |              | Yes              | Yes              | 46.56       | DSN |

---

### Official Review · Reviewer_ai52 · 2023-07-07

**Soundness:** 3 good
**Presentation:** 3 good
**Contribution:** 3 good
**Rating:** 4
**Confidence:** 4

**Summary:**

This paper focuses on task continual learning and attempts to enhance the elastic knowledge transfer across the tasks that sequentially arrive. With the help of masks, achieve forward and backward knowledge transfer.

**Strengths:**

This paper is well written and easy to follow. The proposed method can achieve backword knowledge transfer with the help of mask similarity.

**Weaknesses:**

1. Why can masks identify the similarity of tasks? The mask of the new task before learning should not be determined, so how to complete knowledge transfer at beginning.
2. During the forward transfer, the parameters of the old task were fixed to prevent catastrophic forgetting, while during the reverse transfer, the parameters of the old task were optimized. The two seem somewhat contradictory. Can you unify the two to improve learning efficiency?
3. The EWC and IMM algorithms used in the experiment are both class incremental learning. How can they be compared to Task incremental learning?
4. The operation of convolutional layers is somewhat different from fully connected, and the paper lacks effective descriptions of other structures. Does the convolutional part regards each convolutional kernel as fully connected int the experiments?

**Questions:**

Please refer to the Weakness part.

**Limitations:**

Yes.

---

> ### Author Rebuttal · Authors · 2023-08-09
>
> ## 1. Response to "*mask & knowledge transfer*"
>
> *(1) Forward Knowledge transfer*: Thanks for this question. Our solution is to use the mask mechanism to determine the subnetwork architecture of each arrived task from the hypernetwork $\mathcal{H}$. Thus, any subnetwork is a subset of $\mathcal{H}$. For a newly coming task $t$, the neuron-wise masks regarding $t$ are dynamically changed during the training process, as they are conditioned on the layer embeddings (cf. Eq.(3)). In other words, the embeddings corresponding to the masks are also trained during the task training process, and the final masks for the task $t$ are determined only when the training performance is optimal. As defined in Eq.(5), DSN using mask operation will only update the parameters that are not used in the previous tasks. And we regard the used neurons in the previous tasks as the synapses that only take the role of sending messages between different layers. Hence, the reused neurons from the past tasks allow the forward knowledge transfer to the new task.
>
> *(2) similarity tasks*: Due to the unavailability of old samples, we cannot measure task similarity using the approach of [1], which relies on the distribution of data from previous tasks. In our solution, we treat the subnetwork determined by mask mechanism as knowledge rather than data. Since the subnetworks all originate from the same hypernetwork, with the optimization of Eq. (8), we enforce the new task to reuse more old neurons. Therefore, we measure whether two tasks are similar or not by mask similarity, which is actually subnetwork structure similarity.
>
> [1] Ke Z, Liu B, Huang X. Continual learning of a mixed sequence of similar and dissimilar tasks. NIPS, 2020.
> ## 2. Response to "*backward knowledge transfer*"
>
> Sorry for this confusion. Above we have explained the procedure for forward knowledge transfer. Please allow me to explain the backward knowledge transfer here. For a newly coming task $t$, we will obtain its optimal masks after training convergence. As we claimed that the newly learned knowledge could be beneficial for the past tasks. Thus, in backward knowledge transfer, we first make the task similarity measurement by mask measurement (cf. Eq.(10)) to find the most similar old task (i.e., task argmax($S_t$)). Then, we employ our data-free replay to generate impression crafts. As claimed in the paper, these impressions are to determine whether we need to adjust the subnetwork architecture of the most similar task. During backward knowledge transfer, we will not change any parameters in the hypernetwork as it could cause interference with other tasks. But, we can update the masks of old task argmax($S_t$) as well as the task-specific classifier. That is to say, the subnetwork architecture regarding task argmax($S_t$) can be adjusted while keeping the parameters fixed. In this way, this operation will not cause interference with other old tasks as well as the newly coming task. Based on this interference concern, we cannot unify the two knowledge transfer procedures in our solution due to the sequential nature of the learning process.
>
> ## 3. Response to "*EWC & IMM*"
>
> Thanks for this concern. EWC[1] and IMM[2] have been widely adopted as the baselines for Task incremental learning such as [3][4][5].
>
> [1] Kirkpatrick, James, Razvan Pascanu, Neil Rabinowitz, Joel Veness, Guillaume Desjardins, Andrei A. Rusu, Kieran Milan et al. "Overcoming catastrophic forgetting in neural networks." *Proceedings of the national academy of sciences* 114, no. 13 (2017): 3521-3526.
>
> [2] Lee, Sang-Woo, Jin-Hwa Kim, Jaehyun Jun, Jung-Woo Ha, and Byoung-Tak Zhang. "Overcoming catastrophic forgetting by incremental moment matching." *Advances in neural information processing systems* 30 (2017).
>
> [3] Serra, Joan, Didac Suris, Marius Miron, and Alexandros Karatzoglou. "Overcoming catastrophic forgetting with hard attention to the task." In *International conference on machine learning*, pp. 4548-4557. PMLR, 2018.
>
> [4] Qin, Qi, Wenpeng Hu, Han Peng, Dongyan Zhao, and Bing Liu. "Bns: Building network structures dynamically for continual learning." *Advances in Neural Information Processing Systems* 34 (2021): 20608-20620.
>
> [5] Masana, Marc, Tinne Tuytelaars, and Joost Van de Weijer. "Ternary feature masks: zero-forgetting for task-incremental learning." In *Proceedings of the IEEE/CVF conference on computer vision and pattern recognition*, pp. 3570-3579. 2021.
>
> ## 4. Response to "*convolutional layers*"
>
> Thanks for this concern. We depicted the complete architectures of CNN in Appendix C.3. Following HAT and WSN, our CNN is a modified version of AlexNet containing three convolution layers and two fully-connected layers. Indeed, the convolution layers are different from MLPs. In our experiments, we treat each filter in convolution layers like the 'neuron' in MLPs. And using the mask operation to determine which filter will be activated for the newly coming task training.

---

### Official Review · Reviewer_iEXK · 2023-07-07

**Soundness:** 2 fair
**Presentation:** 2 fair
**Contribution:** 3 good
**Rating:** 5
**Confidence:** 3

**Summary:**

Building upon the principles of the Lottery Ticket Hypothesis, this paper introduces a hyper network model embedded with a series of competitive "ticket" sub-networks. Each of these sub-networks is designed to excel at their corresponding tasks, with a particular emphasis on knowledge transfer. Furthermore, this model promotes not just forward knowledge transfer, but also supports backward transfer of knowledge.

**Strengths:**

Pros:
1. The DSN method facilitates not only forward knowledge transfer but also backward transfer, offering a more dynamic and comprehensive learning paradigm that closely mimics human cognitive processes.

2. The DSN method uses a neuron-wise mask and data-free memory replay, which can significantly save computational resources compared to maintaining a binary mask equal to the number of parameters.

3. By using the mask to select neurons that have been used in earlier tasks and keeping their corresponding weights unchanged, the DSN method effectively addresses the issue of catastrophic forgetting that is common in continual learning scenarios.

4. The data-free replay mechanism treats the trained subnetwork as a past experience and uses it to craft impressions of past samples, which bypasses the need to retain actual past samples, thus addressing privacy concerns and computational overhead.



**Weaknesses:**

Cons:

1. The DSN method may involve complex processes and computations, such as measuring mask similarity scores and fine-tuning the most similar tasks, which might make the method computationally intensive and challenging to implement.

2. While task-specific masks help in achieving better model performance, they add another layer of complexity to the training process and may lead to overfitting if not handled correctly.


**Questions:**

N/A

---

> ### Author Rebuttal · Authors · 2023-08-09
>
> ## 1. Response to "*complex processes*"
>
> *(1) Similarity measurement*: Thanks for this concern. First, the mask similarity measurement is a simple vector-based computation that uses the cosine distance to obtain the similarity scores between different tasks. Second, our masks are neuron-level, which will further decrease the computation cost. For instance, given a task network containing a two-layer MLP and a classifier, each layer consists of $n$ neurons, yielding the neuron mask length for each layer is $n$. Thus, the measuring complexity between two tasks is $\mathcal{O}(n)$ due to the element-wise computation. Instead, the complexity of the existing weight-wise mask mechanism will result in $\mathcal{O}(n^2)$.
>
> *(2) Fine-tune the most similar task*: Thanks for this concern. First, the most important contribution of DSN is to enable knowledge transfer from the current task to the previous task (i.e., backward knowledge transfer). The main additional computation costs happen in this procedure due to the impression crafting and the most similar task fine-tuning. In practice, the complexity of the backward knowledge transfer has been considered during the model design investigation. Specifically, transferring newly learned knowledge to multiple similar old tasks can result in significant time costs (as well as memory costs) that outweigh the benefits. Thus, as we claimed in the paper, we only make backward knowledge transfer to the most similar task. According to Table 2, we can find that our proposed DSN does not cause a huge time consumption. Correspondingly, how to further compress the training time is the focus of our future research. In addition, compared to more recent state-of-the-art methods such as WSN, our number of parameters is significantly less than theirs, making the network more lightweight.
>
> *(3) Implementations*: In our Supplementary Material, our source codes were available and the implementation details were provided in the Appendix.pdf file.
>
> ## 2. Response to "*masks*"
>
> Thanks for this important concern. Several previous studies such as HAT and WSN demonstrate that the mask mechanism can be used to determine a subnetwork for each task from a large hypernetwork. Motivated by achievements in network pruning and Lottery Ticket Hypothesis, a hypernetwork is usually over-parameterized and usually even brings negative impacts on task performance than a smaller network. That is to say, many neurons or weights do not make any contribution to the task performance. In a continual learning context, ours and previous studies such as WSN consider that using the mask can discover a more compact subnetwork for each task. To bypass occasionality and randomness such as overfitting, our main experiments were compared by averaging the results of multiple training rounds (cf. Table 1).

---

> > ### Comment · Reviewer_iEXK · 2023-08-15
> >
> > Thanks a lot for the authors' response, but I still got a few concerns as follows:
> >
> > 1. **Complex Processes**:
> >    - **Similarity Measurement**:
> >      - While it's acknowledged that the mask similarity measurement is a simple vector-based computation using cosine distance, the effectiveness of cosine similarity in capturing true mask similarity remains a concern. This is especially true given the potential high dimensionality of some neural networks.
> >
> >    - **Fine-tune the Most Similar Task**:
> >      - The rationale for performing backward knowledge transfer to only the most similar task is understood, but it does prompt further inquiry. For instance, how do you ensure that the most similar task is the most beneficial for knowledge transfer? Transferring to only one task might miss out on critical shared knowledge between multiple tasks.
> >
> >
> > 2. **Masks**:
> >    - The introduction of mask mechanisms, inspired by other studies such as HAT and WSN, and its relation to the Lottery Ticket Hypothesis is intriguing. However, a key point of contention here is the assumption that a hypernetwork is usually over-parameterized. This assumes a universal pattern across different tasks and datasets, which might not be the case.
> >
> >    - The use of averaging results across multiple training rounds to mitigate concerns like overfitting is commendable. Nevertheless, it would be valuable to see a more comprehensive investigation into how these masks affect model robustness, especially when introduced to out-of-sample or adversarial data.
> >
> > **Conclusion**:
> > While the proposed DSN method is no doubt innovative and offers promise, there remain critical considerations that need addressing for it to be widely accepted and implemented in diverse scenarios.

---

> > > ### Author Response · Authors · 2023-08-15
> > >
> > > Thanks for the comments and valuable time.
> > >
> > > **#Similarity Measurement**
> > >
> > > As each mask correspond to the nueron in the task network, using mask measurement is a vector-level while the weight-based measurement is a tensor-level. In highly dimensional context, in addition to the network scale, there is no doubt that weight-based similarity measurements and even similarity measurements based on task data are affected by the input dimensions, while the mask measurement complexity is only related to the task network scale.
> > >
> > > We also conduct experiment to examine the effect of our similarity measurement. By following [1], we obtain the task similarity between two tasks using their real samples. And we obtain the mask similarity in our DSN. Fig. 1 shows the similarity results on PMNIST after training task 10. We can observe that our mask similarity has a similar function to task similarity in [1]. However, as we claimed before, we cannot use the task similarity directly due to the unavailability of old samples. We should have submitted a pdf conains visualized figure, however, we miss the deadline. Sorry for our ignorance.
> > >
> > > |     | Mask Similarity(%) | Task Similarity(%) |
> > > | --- | --- | --- |
> > > | Task 1 | 46.39 | 45.92 |
> > > | Task 2 | 53.26 | 49.32 |
> > > | Task 3 | 57.49 | 54.43 |
> > > | Task 4 | 63.94 | 59.68 |
> > > | Task 5 | 69.42 | 66.05 |
> > > | Task 6 | 72.06 | 75.69 |
> > > | Task 7 | 73.41 | 72.21 |
> > > | Task 8 | 74.75 | 78.79 |
> > > | Task 9 | 78.39 | 80.15 |
> > >
> > > [1] Ke Z, Liu B, Huang X. Continual learning of a mixed sequence of similar and dissimilar tasks[J]. Advances in Neural Information Processing Systems, 2020, 33: 18493-18504.
> > >
> > > **#Mask**
> > >
> > > In response1 of bsXa, we have provided additional experiments. We consider that transferring new knowledge to multiple old tasks can result in significant time costs (as well as memory costs) that outweigh the benefits. Thus, we only make backward knowledge transfer to the most similar task. Herein, we also provided experiments that DSN can transfer to more old tasks as below. We find that this version can further promote model performance. However, the time cost is significantly increased.

---

### Decision · Program_Chairs · 2023-09-21

**Decision:**

Accept (poster)

**Comment:**

This paper proposes a new approach to task-incremental learning called "Data-free Subnetwork" (DSN) which is a masking based method. The paper also proposes a data-free replay method to update the masks of previous tasks after learning new tasks. This enables backward transfer.

There were a lot of questions from the reviewers about the methodology. Authors clarified everything in the rebuttal. I do not think the authors need to apply their algorithm to class-incremental learning.

Given there is no major concern about the work, even from the low rating reviewers, I recommend an accept.

I recommend the authors to incorporate all their explanations and additional experiments into the final version of the paper.